# PixWizard: Versatile Image-to-Image Visual Assistant with Open-Language Instructions

**Weifeng Lin[1]*   Xinyu Wei[2]*   Renrui Zhang[1]*   Le Zhuo[1,3]   Shitian Zhao[3]**

**Siyuan Huang[3]   Junlin Xie[3]   Peng Gao[3]‡   Hongsheng Li[1]†**

[1]CUHK MMLab    [2]Peking University    [3]Shanghai AI Laboratory

## Abstract

This paper presents a versatile image-to-image visual assistant, **PixWizard**, designed for image generation, manipulation, and translation based on free-from language instructions. To this end, we tackle a variety of vision tasks into a unified image-text-to-image generation framework and curate an *Omni Pixel-to-Pixel Instruction-Tuning Dataset*. By constructing detailed instruction templates in natural language, we comprehensively include a large set of diverse vision tasks such as text-to-image generation, image restoration, image grounding, dense image prediction, image editing, controllable generation, inpainting/outpainting, and more. Furthermore, we adopt Diffusion Transformers (DiT) as our foundation model and extend its capabilities with a flexible any resolution mechanism, enabling the model to dynamically process images based on the aspect ratio of the input, closely aligning with human perceptual processes. The model also incorporates structure-aware and semantic-aware guidance to facilitate effective fusion of information from the input image. Our experiments demonstrate that PixWizard not only shows impressive generative and understanding abilities for images with diverse resolutions but also exhibits generalization capabilities with unseen tasks and human instructions.

## 1 Introduction

Large Language Models (LLMs) (Brown et al., 2020; Touvron et al., 2023) and Large Vision Models (LVMs) (Radford et al., 2021; Caron et al., 2021; Bai et al., 2024) have gained global popularity by successfully unifying multiple tasks within a single, coherent framework. Nowadays, LLMs have become efficient language assistants, demonstrating strong capabilities in open-world language understanding and reasoning. However, a versatile visual assistant capable of following diverse multimodal instructions that align with human intentions to effectively perform various visual tasks in real-world scenarios is still under exploration.

Recently, there are two research lines aiming to achieve general visual assistants: diffusion-based and in-context learning approaches. The first focuses on developing text-to-image models (Rombach et al., 2022b) as a unified foundation model for various visual perception tasks. For example, InstructPix2Pix (Brooks et al., 2023), InstructDiffusion (Geng et al., 2024), and InstructCV (Gan et al., 2024) repurpose generative models as a universal language interface for image editing and other visual tasks. However, their capable tasks are limited, and their performance lags behind that of task-specific models. The second research direction focuses on visual prompting, where pixel-based prompts are used to tackle various vision tasks within a single learning framework. This approach employs prompting techniques to generate the desired visual outputs in an in-context manner. Examples include Painter (Wang et al., 2023b), PromptDiffusion (Wang et al., 2023c), and LVM (Bai et al., 2024), which have successfully handled a variety of visual scenarios within one framework. However, these methods inherently lack the ability to follow human language instructions, limiting their controllability and interactivity. (More related works are discussed in Sec. A.)

---

*Equal Contribution

†Corresponding Authors

‡Project Lead

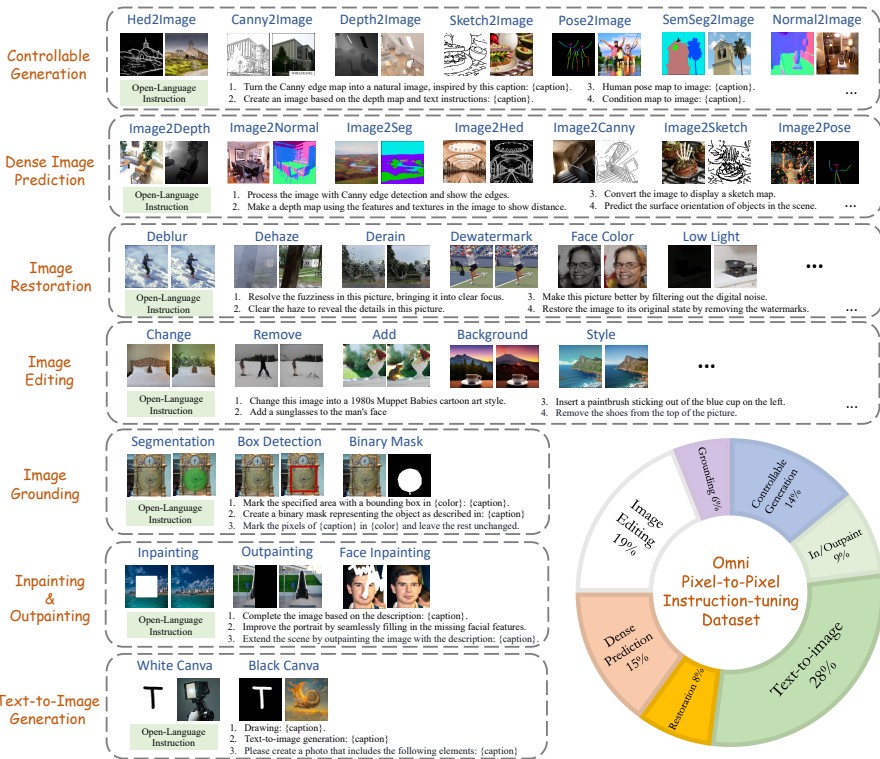

Figure 1: Task Overview of the Omni Pixel-to-Pixel Instruction-tuning Dataset for PixWizard.

Taking into account the strengths and limitations of previous approaches, we introduce **PixWizard**, a versatile interactive image-to-image visual assistant designed for image generation, manipulation, and image-to-image translation. PixWizard is a DiT-based framework that can handle a wide range of visual tasks when provided with sufficient training data, along with the interface for human language instructions. Specifically, PixWizard exhibits three main features as follows:

• *1. Task Unification:* Given the diverse nature of vision tasks and data formats, ranging from pixels and coordinates to binary masks and categories, it is challenging to find a unified representation. We observe that most vision tasks can be framed as image-to-image translation problems. For tasks not naturally suited to image output, we first learn to generate their visualizations and then apply post-processing to convert them into the desired formats. This approach is a key step toward developing a versatile visual assistant.

• *2. Data Construction:* We aim to leverage and integrate the remarkable diversity of tasks and data in the visual domain. To achieve this, we have built a comprehensive training set with a total of 30 million data points, enabling our model to support five main capabilities: *(i)* Image generation, which includes text-to-image generation, controllable generation, inpainting, and outpainting; *(ii)* Image editing; *(iii)* Image restoration, covering tasks such as deraining, desnowing, deblurring, super-resolution, and more; *(iv)* Image grounding, which involves locating objects based on user prompts; and *(v)* Dense image prediction, which includes depth estimation, surface normal estimation, pose estimation, semantic segmentation, and image-to-canny/HED/sketch/Line-Art conversions.

• *3. Architecture Design:* For a robust visual assistant, the architecture and scalability of the foundation model are crucial. In this work, we use the flow-based Diffusion Transformer (DiT) (Ma et al., 2024), which offer great versatility and stability in modeling data distributions by learning conditional velocity fields. The DiT architecture further enhances scalability and is well-suited for incorporating conditional information. Building on this foundation, we introduce several key components. We extend the dynamic partitioning and padding scheme (Zhuo et al., 2024) to handle input images of any resolution, aligning closely with human perception. Additionally, we implement structure-aware and semantic-aware guidance, enabling the model to follow multimodal instructions (images and user prompts) for a wide range of manipulations.

Experimental results show that PixWizard achieves competitive performance on certain tasks compared to task-specific vision models, and outperforms current state-of-the-art general visual models

in most tasks, delivering its strong overall multi-task performance. More importantly, our model handles tasks and instruction prompts it has not encountered during training, demonstrating promising generalization capabilities. This further highlights PixWizard's strength as a powerful interactive image-to-image visual assistant.

## 2 OMNI PIXEL-TO-PIXEL INSTRUCTION-TUNING DATASET

To equip our image-to-image visual assistant with comprehensive capabilities in image generation, manipulation, and translation, we compiled a multi-task training dataset for visual instruction tuning, consisting of 30 million instances across seven primary domains, as illustrated in Fig. 1. This is the user-friendly image-instruction-image triplet dataset, built from both open-source and in-house data, filtered with the help of MLLMs and manual review.

**Image Restoration.** We incorporate low-level data to restore images degraded by various environmental or technical factors. This section utilizes a wide range of open-source datasets covering key restoration tasks, including *(1) Denoising*, *(2) Deraining*, *(3) Demoireing*, *(4) Dehazing*, *(5) Deblurring*, *(6) Desnowing*, *(7) Deshadowing*, *(8) Low-Light Enhancement*, *(9) Face Restoration*, *(10) Watermark Removal*, and *(11) Super Resolution*. Since both the inputs and outputs are inherently defined in the RGB space, these tasks can be seamlessly unified by our PixWizard model without any extra transformations. All open-source datasets we use are provided in Sec. B.1.

**Image Grounding.** Image grounding involves identifying and highlighting specific areas of objects in images based on provided text prompts. The data for this part is sourced from well-known datasets such as gRefCOCO (Liu et al., 2023a), RefCOCO3 (Yu et al., 2016), and Visual Genome (Krishna et al., 2017). We focus on three types of grounding tasks: *(1) Segmentation Referring*, where the target object specified by the user is highlighted in the output image; *(2) Box Detection Referring*, where the target object is highlighted using bounding boxes; and *(3) Binary Mask Prediction*, where a binary mask is directly predicted. (Details are provided in the Sec. B.2.)

**Controllable Generation.** Referring to ControlNet (Zhang et al., 2023a), we aim to equip our model with natural image generation capabilities given conditional inputs. We first gather natural images from the LAION Art dataset (Schuhmann et al., 2022) and our own collection of high-quality images from the Internet. We then use MiniCPM-Llama3-V 2.5 (Yao et al., 2024), a robust edge-side multimodal LLM, along with advanced techniques to generate captions and conditional inputs for the images. (Details on data construction can be found in Sec. B.3.)

**Dense Image Prediction.** Dense image prediction tasks require the model to interpret input images and produce dense annotations. For *depth estimation, surface normal estimation, and semantic segmentation*, we represent per-pixel labels as RGB images, which can be generated via the image generation capabilities. For other tasks, such as the prediction of *human pose maps*, *sketches*, *HED boundaries*, *canny edge maps*, and *cartoon line art*, we treat them as image-to-image translation tasks, as we can easily obtain image pairs using open-source tools. (Details are provided in the Sec. B.4.)

**Image Editing.** Instruction-based image editing holds great potential for practical applications, as it allows users to perform edits using natural language commands. To enhance our model's ability to modify images according to specific user instructions, we unify several public editing datasets, including UltraEdit (2024), MagicBrush (2024a), GQA-Inpaint (2023), Instruct P2P (2023), SEED-X-Edit (2024), GIER (2020), and HQ-Edit (2024). These datasets encompass a wide range of semantic entities, varying levels of detail, and multiple editing tasks including object removal, object replacement, object addition, background replacement, and style transfer.

**Inpainting** involves filling in missing parts of an image with new or modified content. To create inpainting instances, we apply random black or white masks to different regions of the original images. These masks come in various shapes, including circles, rectangles, and free-form patterns, and are randomly placed on the images, resulting in a wide range of occluded areas for inpainting. **Outpainting (Image extrapolation)** extends an image's content beyond its boundaries. To create outpainting instances, we randomly crop the central part of the image, while masking the surrounding areas with black or white. The cropping is not limited to square shapes but includes rectangles of varying proportions. This approach ensures that the outpainting task covers a range of extension scenarios, challenging the model to generate coherent content beyond the original image boundaries.

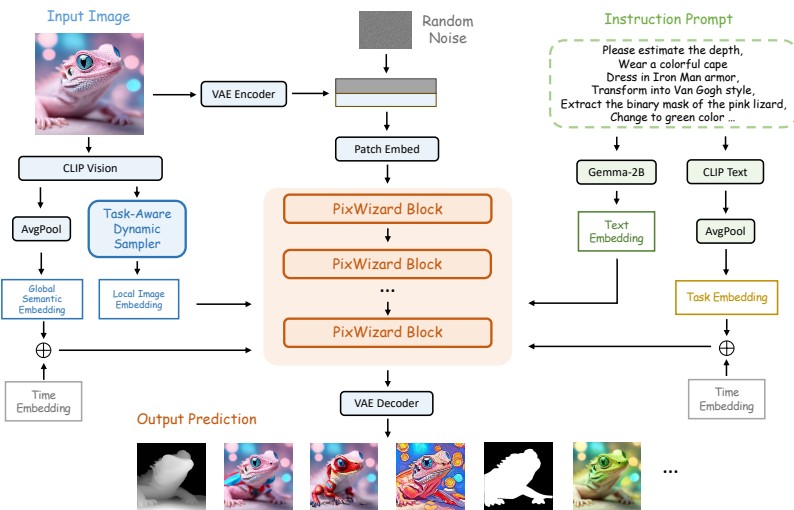

Figure 2: Overall framework of PixWizard.

**Text-to-Image Generation.** Image-to-image and text-to-image are distinct tasks: the former requires an additional input image as a condition, while the latter does not. Existing instruction-tuning methods adapt pretrained text-to-image models for image-to-image tasks, but often lose the original text-to-image capabilities. To unify both tasks within a single framework, we propose a "drawing" strategy that preserves text-to-image functionality. Specifically, we introduce an additional input—a fully white or black image—alongside language instructions. This simulates a blank canvas, allowing the model to "draw" images based on the text prompts. This approach differentiates our model from previous text-to-image systems.

**Open-language Instruction.** To enhance the usability of PixWizard as a practical visual assistant, we aim for the model to understand free-form user prompts. Instead of relying on fixed task-specific prompts, we begin by manually writing 6-10 prompts for each vision task to describe the task. We then use GPT-4o to generate a wide range of paraphrased variations. This process ensures that our instruction set remains diverse while staying true to the core intent of the original prompts. Instruction templates and examples are provided in Sec. B.5.

## 3 PIXWIZARD

In this section, we present the details of the PixWizard framework from the perspectives of model architecture (as shown in Fig. 2) and training strategies.

### 3.1 FLOW-BASED CONDITIONAL INSTRUCTION-TUNING

Previous studies (Wang et al., 2022b; Brooks et al., 2023) show that fine-tuning large diffusion models outperforms training models from scratch for image translation and editing tasks. Therefore, we initialize the weights of PixWizard with the pretrained Lumina-Next-T2I (Zhuo et al., 2024) checkpoint, which is a flow-based diffusion transformer, to leverage its extensive text-to-image generation capabilities. We learn a network $v_\theta$ that predicts the velocity field $u_t$ given image conditioning $c_I$ and text instruction conditioning $c_T$. We minimize the loss function as follow:

$$\mathcal{L} = \mathbb{E}_{t,p_1(x_1),p_t(x_t|x_1),c_I,c_T} \|v_\theta(x_t, t, c_I, c_T) - u_t(x_t, t|x_1)\|^2. \tag{1}$$

### 3.2 ARCHITECTURE

**Text Encoders.** We begin by using Gemma-2B (Team et al., 2024) as the text embedder in PixWizard to encode text prompts. However, in multi-task learning, relying solely on text instructions is insufficient to guide the model in accurately executing user commands. To better guide the generation process for the correct task, we incorporate the CLIP text encoder (Radford et al., 2021). Global average pooling is applied to the CLIP text embeddings to obtain a coarse-grained text representation, which is passed through an MLP-based task embedder to generate the task embedding.

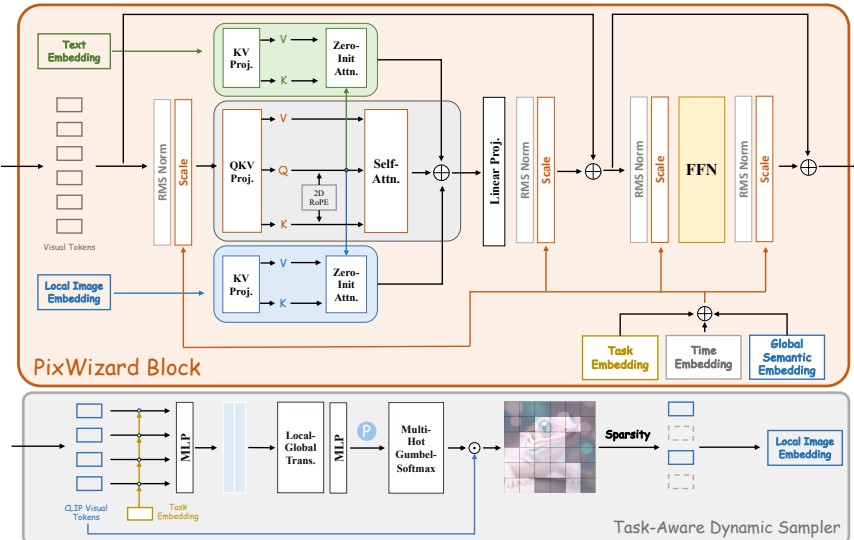

Figure 3: The schematic illustrations of PixWizard Block and Task-Aware Dynamic Sampler.

This embedding is then integrated into the PixWizard Block by adding it to the timestep embeddings through a modulation mechanism. In Sec. C.2, we show the t-SNE visualization to further illustrate the effectiveness of the task embedding.

**Structural-Aware Guidance.** To effectively capture the overall structural features of the input image condition, we begin by encoding the image using a variational autoencoder (VAE) (Kingma & Welling, 2013) from SDXL (Podell et al., 2023). Next, we concatenate the image latent with the noise latent along the channel dimension (Saharia et al., 2022c;a). Following (Brooks et al., 2023), additional input channels are added to the patch embedder, with the weights for these new channels initially set to zero.

**Semantic-Aware Guidance.** Besides recognizing structural features, we use CLIP L/14-336 (Radford et al., 2021) to obtain semantic image embeddings. Within the PixWizard block, we introduce two zero-initialized attention mechanisms, allowing latent target image tokens to query information from the condition keys and values. Specifically, a zero-initialized gating mechanism is employed to gradually inject conditional image and text information into the token sequences. Given target image queries ($Q_i$), keys ($K_i$), and values ($V_i$), along with text instruction keys ($K_t$) and values ($V_t$), and conditional image keys ($K_{ci}$) and values ($V_{ci}$), the final attention output is formulated as:

$$A = \text{softmax}\left(\frac{\tilde{Q}_i \tilde{K}_i^T}{\sqrt{d}}\right) V_i + \tanh(\alpha_t) \text{softmax}\left(\frac{\tilde{Q}_i K_t^T}{\sqrt{d}}\right) V_t + \tanh(\alpha_{ci}) \text{softmax}\left(\frac{\tilde{Q}_i K_{ci}^T}{\sqrt{d}}\right) V_{ci},$$

(2)

where $\tilde{Q}_i$ and $\tilde{K}_i$ stand for applying RoPE (Su et al., 2024), $d$ is the dimension of queries and keys, and $\alpha$ indicates the zero-initialized learnable parameter in gated cross-attention.

However, inputting all image tokens into the attention layer can lead to significant computational demands, and not all semantic tokens are relevant to the specific task. To address this, we introduce the Task-Aware Dynamic Sampler, designed to select the most relevant semantic tokens for each task. This sampler uses a lightweight ranking network with four linear layers and activation functions. Inspired by DynamicViT (Rao et al., 2021), we map image tokens to both local and global features. Additionally, task embeddings ($x_{task}$) are integrated to help the sampler identify the tokens most relevant to the task. The computational process is formulated as:

$$z = MLP(x + x_{task}) \in \mathbb{R}^{N \times C},$$

(3)

$$z^{\text{local}} = z[:, : \frac{C}{2}] \in \mathbb{R}^{N \times \frac{C}{2}}, z^{\text{global}} = Avg(z[:, \frac{C}{2} :]) \in \mathbb{R}^{1 \times \frac{C}{2}}, z'_i = [z_i^{\text{local}}, z^{\text{global}}], \quad 1 \leq i \leq N,$$

(4)

$$M = MLP(z') \in \mathbb{R}^{N \times 1}$$

(5)

where $M_i$ denotes the importance of the $i$-th token. However, implementing token sparsification is challenging in practice. Directly sampling tokens based on their importance scores is non-differentiable, which hinders end-to-end training. To address this, we use the Gumbel-Softmax

technique (Jang et al., 2016) and adapt it into a Multi-Hot Gumbel-Softmax (MHGS) to enable simultaneous sampling of the top $K$ tokens:

$$\hat{x} = x \odot MHGS(M) \tag{6}$$

where the output of GumbelSoftmax is a multi-hot tensor representing the mask of the retained tokens. $\odot$ denotes the Hadamard product, where the top $K$ tokens by importance score are weighted by 1 and retained, while the remaining $(N - K)$ tokens are weighted by zero and discarded. Each layer is equipped with an independent task-aware dynamic sampler. This approach not only captures the most relevant semantic features needed by each layer for different tasks but also reduces the computational cost in the attention process.

### 3.3 TWO-STAGE TRAINING AND DATA BALANCING STRATEGIES

To unlock the model's potential and improve its performance on tasks with smaller datasets, we propose a two-stage training and data-balancing strategy. **S1:** In the first stage, we initialize the model by combining the weights of a pre-trained Lumina-Next-T2I (Zhuo et al., 2024) with randomly initialized weights for the newly added modules. We prioritize tasks with smaller datasets, assigning each a sampling weight to increase its data volume. This weight determines how many times the dataset is repeated during an epoch. Using this method, each task achieves approximately 20k data points. We then randomly sample from other tasks to match this scale, creating our first-stage training dataset, with training spanning 4 epochs. **S2:** In the second stage, we initialize the model using the weights from the first stage and combine all the collected data for further training. To balance tasks, we assign manual sampling weights to each dataset, randomly selecting data when a weight is less than 1.0. We also include text-to-image data at a 1:1 ratio with other tasks, resulting in a second-stage training dataset. At this stage, the total dataset reaches up to 20 million samples.

## 4 EXPERIMENTS

### 4.1 FIRST SECTION RESULTS

**Settings.** For image restoration, we follow previous works (Conde et al., 2024; Potlapalli et al., 2024) and prepare datasets for various restoration tasks during training. For evaluation, we first select two representative benchmarks: Rain100L (2017) for deraining and SIDD (2018) for denoising. Additionally, we further assess performance on other restoration tasks and evaluate zero-shot capabilities in the Sec. D.2.

For image grounding, we evaluate referring segmentation tasks on the gRefCOCO (2023a), RefCOCO, and RefCOCO+ validation and test sets. To assess the performance gap with specialized models, we report results from several expert methods and primarily compare our approach with two unified models: Unified-IO and InstructDiffusion. Following standard practices (Liu et al., 2023a), we use cumulative IoU (cIoU) as the performance metric.

Table 1: Comparison of PixWizard with task-specific and vision generalist baselines across six representative tasks, covering both high-level visual understanding and low-level image processing. '×' indicates that the method is incapable of performing the task.

| Methods | Depth Est. RMSE↓ | | Semantic Seg. mIoU↑ | Surface Normal Est. Mean Angle Error↓ | Denoise PSNR↑ SSIM↑ | | Derain PSNR↑ SSIM↑ | | Image Grounding cIoU↑ (val set) | |
|---|---|---|---|---|---|---|---|---|---|---|
| | NYUv2 | SUNRGB-D | ADE20K | NYU-Depth V2 | SIDD | | Rain100L | | RefCOCO | RefCOCO+ |
| DepthAnything (2024a) | 0.206 | - | | | | | | | | |
| Marigold (2024a) | 0.224 | - | | | | | | | | |
| Mask DINO (2023) | | | 60.80 | | | | | | | |
| Mask2Former (2022) | | | 56.10 | | | | | | | |
| Bae et al. (2021) | | | | 14.90 | | | | | | |
| InvPT (2022) | | | | 19.04 | | | | | | |
| AirNet (2022) | | | | | 38.32 | 0.945 | 32.98 | 0.951 | | |
| PromptIR (2024) | | | | | 39.52 | 0.954 | 36.37 | 0.972 | | |
| LAVT (2022) | | | | | | | | | 72.73 | 56.86 |
| ReLA (2023a) | | | | | | | | | 73.21 | 56.10 |
| Unified-IO (2022b) | 0.387 | 0.287 | 25.71 | - | × | × | × | × | 46.42 | **40.50** |
| Painter (2023b) | 0.288 | 0.285 | **49.90** | × | **38.71** | 0.954 | 29.87 | 0.882 | × | × |
| InstructCV (2024) | 0.297 | **0.279** | 47.23 | × | × | × | × | × | × | × |
| InstructDiffusion (2024) | × | × | × | × | 34.26 | 0.938 | 19.82 | 0.741 | 41.64* | 33.20* |
| PixWizard | **0.287** | 0.291 | 32.76 | **19.65** | 38.67 | **0.957** | **31.43** | **0.917** | **46.44** | 36.49 |

Dense image prediction tasks are evaluated across three vision tasks: ADE20k (2017b) for semantic segmentation, NYUv2 (2012) and SUNRGB-D (2014) for monocular depth estimation, and NYU-Depth v2 (2012) for surface normal estimation. Implementation details can be found in Sec. D.1.

**Results.** Table 1 presents a comprehensive performance comparison with recent state-of-the-art (SOTA) task-specific and all-in-one methods. As shown in the results, despite denoising and deraining data making up only a small portion of the overall training set, our method outperforms other unified methods and compare to some task-specific approaches. In the image grounding task, PixWizard significantly outperforms the diffusion-based generalist model InstructDiffusion by 4.8 cIoU on RefCOCO (val). However, there is still room for improvement compared to other highly specialized models. Furthermore, as shown in Fig.5, PixWizard supports flexible instructions, allowing it to highlight and visualize the target object directly on the image while also generating the corresponding binary mask. Additional quantitative evaluation results are provided in the Sec. D.3.

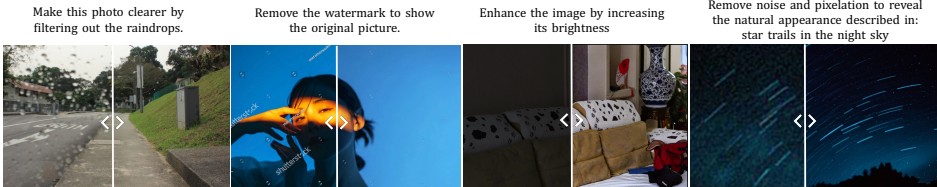

Figure 4: Qualitative Evaluation of Instruction-based Image Restoration.

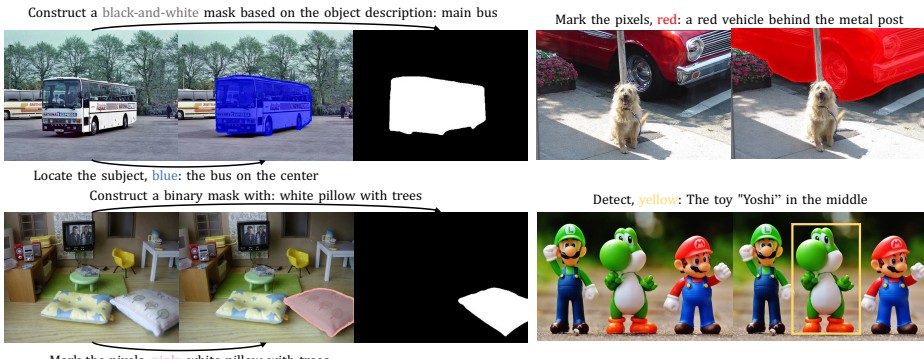

Figure 5: Qualitative Results of Instruction-based Image Grounding.

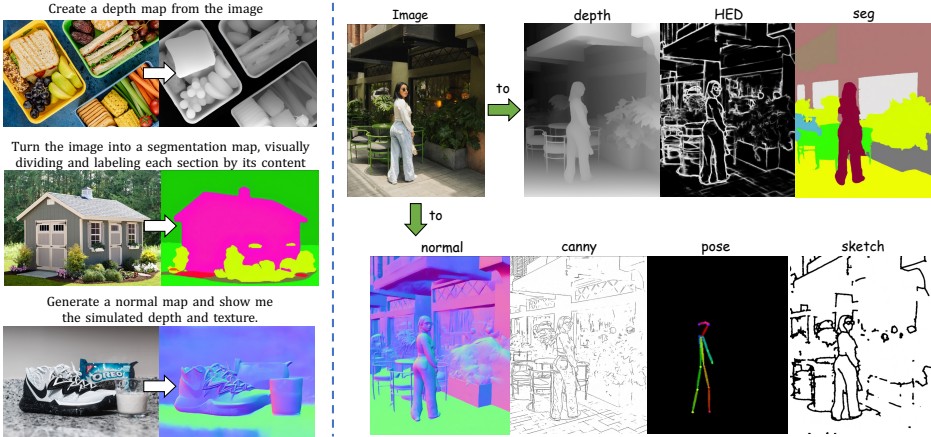

Figure 6: Visualizations of dense image prediction examples.

For dense prediction tasks, PixWizard performs competitively against both generalist and task-specific baselines across all three tasks. In depth estimation on the NYUv2 test set, PixWizard achieves a 10.0% improvement in RMSE compared to Unified-IO and performs similarly to Painter and InstructCV. Additionally, Fig. 6 provides examples of PixWizard's outputs. As shown, by supplying the corresponding task-specific prompt for the same image, we can easily generate the respective condition visualization, underscoring PixWizard's significant practical value.

## 4.2 SECOND SECTION RESULTS (IMAGE EDITING)

**Settings.** We evaluate PixWizard across two benchmarks, the MagicBrush Test (Zhang et al., 2024a) and the Emu Edit Test (Sheynin et al., 2024), to assess its effectiveness in image editing capabilities. For a fair comparison, we primarily compare it with several instruction-guided image editing methods. Consistent with Emu Edit, we use L1 distance, CLIP image similarity, DINO similarity, CLIP text-image similarity, and CLIP text-image direction similarity as metrics.

Table 2: Comparison with image-editing baselines evaluated on Emu Edit and MagicBrush test set.

| Method | Emu Edit Test set | | | | | MagicBrush Test Set | | | | |
| | $CLIP_{dir}\uparrow$ | $CLIP_{im}\uparrow$ | $CLIP_{out}\uparrow$ | L1↓ | DINO↑ | $CLIP_{dir}\uparrow$ | $CLIP_{im}\uparrow$ | $CLIP_{out}\uparrow$ | L1↓ | DINO↑ |
|---|---|---|---|---|---|---|---|---|---|---|
| InstructPix2Pix (2023) | 0.078 | 0.834 | 0.219 | 0.121 | 0.762 | 0.115 | 0.837 | 0.245 | 0.093 | 0.767 |
| MagicBrush (2024a) | 0.090 | 0.838 | 0.222 | 0.100 | 0.776 | 0.123 | 0.883 | 0.261 | 0.058 | 0.871 |
| Emu Edit (2024) | **0.109** | **0.859** | 0.231 | 0.094 | **0.819** | **0.135** | **0.897** | 0.261 | **0.052** | **0.879** |
| UltraEdit (2024) | 0.107 | 0.844 | **0.283** | 0.071 | 0.793 | - | 0.868 | - | 0.088 | 0.792 |
| PixWizard | 0.104 | 0.845 | 0.248 | **0.069** | 0.798 | 0.124 | 0.884 | **0.265** | 0.063 | 0.876 |

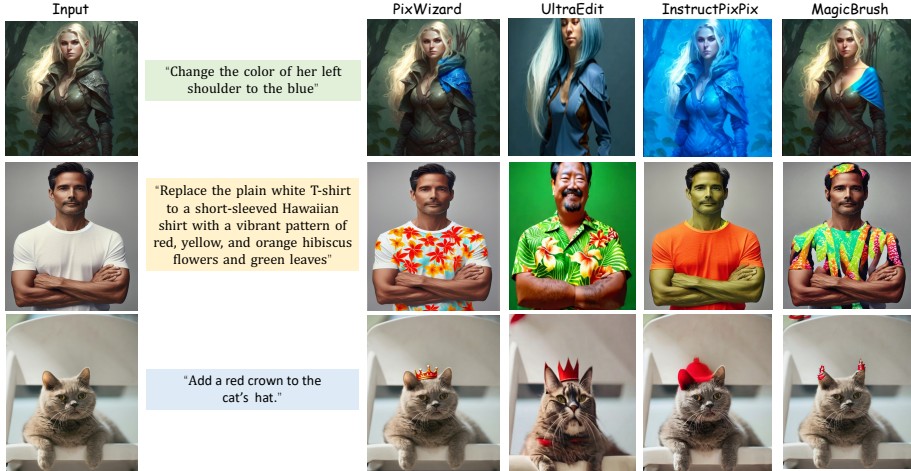

Figure 7: Qualitative examples comparing PixWizard with other editing approaches.

**Results.** Table 2 presents our results compared to the baselines. The findings show that our model consistently outperforms InstructPix2Pix, MagicBrush, and UltraEdit in automatic metrics and achieves comparable performance to state-of-the-art method Emu Edit. As shown in Fig. 7, our model precisely identifies the editing region while preserving the rest of the pixels, and it demonstrates the best understanding of the given instructions.

## 4.3 THIRD SECTION RESULTS (IMAGE GENERATION)

In this section, we focus on evaluating the effectiveness of PixWizard's generation capabilities. Specifically, we assess its performance across four tasks: text-to-image generation, controllable image generation, inpainting, and outpainting. Implementation details can be found in Sec. D.1.

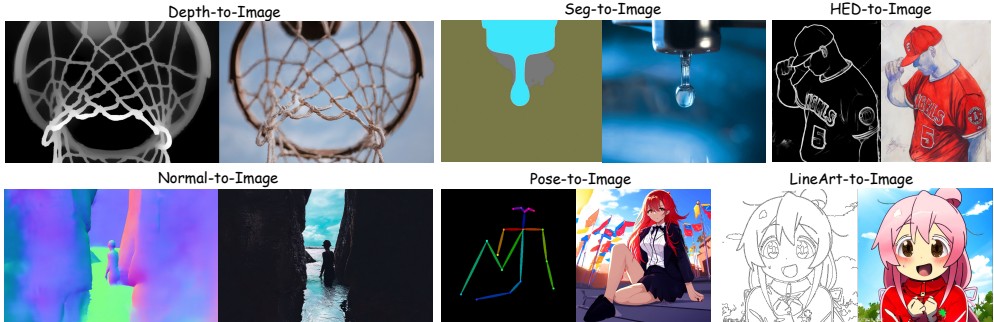

Figure 8: Visualization examples under different conditions.

Table 3: Comparison of PixWizard with task-specific baselines across five representative tasks.

| Methods | Canny-to-Image | | | Depth-to-Image | | | Inpainting | | Outpainting | | Text-to-Image | |
|---|---|---|---|---|---|---|---|---|---|---|---|---|
| | FI↑ | FID↓ | CLIP-S↑ | RMSE↓ | FID↓ | CLIP-S↑ | FID↓ | LPIPS↓ | FID↓ | IS↑ | FID↓ | HPSv2↑ |
| | MultiGen-20M | | | MultiGen-20M | | | Places | | Places | | COCO-30K | Photo |
| ControlNet-SD1.5 (2023a) | 34.65 | 14.73 | **32.15** | 35.90 | 17.76 | **32.45** | | | | | | |
| T2I-Adapter-SD1.5 (2024) | 23.65 | **15.96** | 31.71 | 48.40 | 22.52 | 31.46 | | | | | | |
| LDM-4 (2022b) | | | | | | | 9.39 | 0.246 | | | | |
| LaMa (2022) | | | | | | | 12.0 | **0.24** | | | | |
| DeepFill v2 (2019) | | | | | | | | | 11.51 | 17.70 | | |
| MaskGIT (2022b) | | | | | | | | | 7.80 | **22.95** | | |
| DALL·E 2 (2021) | | | | | | | | | | | 10.32 | 27.24 ± 0.198 |
| SD 1.5 (2022b) | | | | | | | | | | | 9.62 | 27.46 ± 0.198 |
| PixArt-α (2024b) | | | | | | | | | | | **7.32** | - |
| Lumina-Next (2024) | | | | | | | | | | | 9.79 | 27.47 ± 0.203 |
| PixWizard | **35.46** | 15.76 | 32.01 | **33.83** | **16.94** | 31.84 | **9.27** | 0.25 | **7.54** | 22.18 | 9.56 | **27.47 ± 0.183** |

**Controllable Generation Results.** Without the need for separate training for each model, PixWizard is an all-in-one solution capable of handling multiple conditions. As shown in Table 3 and Fig. 8, PixWizard achieves the highest controllability and best image quality under depth conditions, while also being comparable to current separate models in image-text alignment.

**Inpainting Results.** As shown in Table 3, PixWizard outperforms other inpainting approaches, improving overall image quality based on FID and LPIPS scores. The qualitative examples in Fig. 9 further demonstrate PixWizard's effectiveness in generating coherent content. Building on its strong inpainting capabilities, PixWizard enables users to perform more precise image editing tasks, such as *Remove Anything*, *Replace Anything*, and *Add Anything*. More details can be found in Sec. D.4

**Outpainting Results.** As shown in the quantitative comparison results in Table 3, PixWizard outperforms other baselines in the outpainting task, delivering state-of-the-art image generation quality with a FID score of 7.54 and an IS score of 22.18. The samples in Fig. 9 demonstrate PixWizard's ability to synthesize images in various scenes and styles, and it flexibly handles image extrapolation in multiple directions and aspect ratios with better marginal coherence.

**Text-to-Image Results.** As the results shown in Table 3, PixWizard achieves a FID score of 9.56 when tested for zero-shot performance on the COCO dataset. Although some models achieve a lower FID, they focus solely on text-to-image tasks and rely on significantly more training resources. Additionally, we evaluated the Human Preference Score (HPS v2), a robust benchmark for assessing human preferences in text-to-image synthesis. PixWizard performed well, delivering image quality comparable to popular text-to-image generators. We provide visual samples in Fig. 9. PixWizard supports high-resolution image synthesis, up to $1024 \times 1024$, with any resolution and aspect ratio.

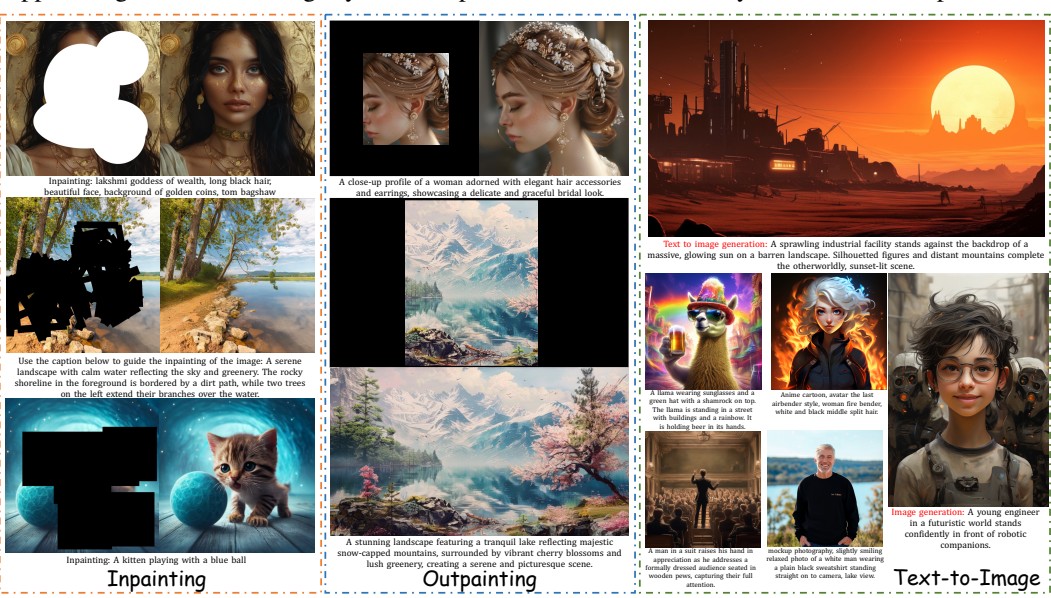

Figure 9: Visualization results of **Inpainting**, **Outpainting** and **Text-to-Image Generation**.

## 4.4 ABLATION STUDY

We performed ablation studies to assess the impact of each component's design and the training process on learning in PixWizard. Given computational limitations, we conducted the ablation on PixWizard, training it for 40k steps.

Table 4: Comparison of the models with two different guidances, dynamic semantic tokens sampling (DSTS), and two-stage training and data balancing strategy for different tasks.

| Methods | Deraining
Rain100L | RefCOCO
val | Depth Estim.
NYUv2 | Image Editing
Emu Edit | Canny-to-Image
MultiGen-20M | | Inpainting
Places |
|---|---|---|---|---|---|---|---|
| | PSNR↑ | cIoU↑ | RMSE↓ | $CLIP_{dir}$↑ | F1↑ | FID↓ | FID↓ |
| M1 | $29.91_{(-0.33)}$ | $40.78_{(-0.94)}$ | $0.319_{(+0.005)}$ | $0.078_{(-0.010)}$ | $32.98_{(-0.03)}$ | $17.41_{(-0.27)}$ | $10.91_{(-0.03)}$ |
| M2 | $14.72_{(-15.52)}$ | $18.43_{(-23.29)}$ | $0.586_{(-0.262)}$ | $0.071_{(-0.017)}$ | $11.12_{(-23.89)}$ | $19.34_{(-2.20)}$ | $13.87_{(-2.99)}$ |
| PixWizard w/o DSTS | $30.19_{(-0.05)}$ | $41.66_{(-0.06)}$ | $0.318_{(+0.006)}$ | $0.091_{(+0.003)}$ | $32.93_{(-0.08)}$ | $17.02_{(+0.12)}$ | $10.93_{(-0.05)}$ |
| PixWizard w/o two-stage | $29.17_{(-1.07)}$ | $40.18_{(-1.54)}$ | $0.322_{(+0.002)}$ | $0.085_{(-0.003)}$ | $32.87_{(-0.14)}$ | $17.21_{(-0.07)}$ | $10.97_{(-0.09)}$ |
| PixWizard | 30.24 | 41.72 | 0.324 | 0.088 | 33.01 | 17.14 | 10.88 |

**Structural-Aware vs. Semantic-Aware Guidance.** We demonstrate the importance of integrating both structure-aware and semantic-aware guidance to enhance PixWizard's performance across diverse tasks. To validate this, we trained two additional models: $M1$ with only the structure-aware module and $M2$ with only the semantic-aware module. Results show that $M1$ performs better on tasks requiring preservation of image structure and details, while $M2$, which relies solely on cross-attention for injecting image features, struggles with most visual tasks and often generates outputs that deviate from the input. However, as noted by other methods (Ye et al., 2023; Hu et al., 2024), semantic guidance excels in tasks needing flexibility, such as text-to-image generation, conditional generation, and image editing. In comparison, PixWizard, which combines both modules, achieves balanced performance across a broader range of tasks.

**Influence of Dynamic Semantic Tokens Sampling.** As shown in Table 4, the impact of Dynamic Semantic Tokens Sampling (DSTS) on task performance is minimal, but it improves overall performance on average. This indicates that modeling semantic tokens for the entire image is unnecessary. By dynamically sampling relevant features for each task's focus, the model operates more efficiently. Moreover, using fewer tokens reduces computational load during attention calculations, resulting in faster inference.

**Influence of Two-Stage Training and Data Balancing.** Table 4 presents the results of our proposed two-stage training and data balancing strategy. As shown, our approach is crucial, as the two-stage method significantly improves performance on tasks with smaller datasets while maintaining the same number of training steps and achieving faster convergence. Notably, for tasks with larger datasets, the performance remains comparable.

## 5 DISCUSSION AND CONCLUSION

In this work, we explore how to build a versatile interactive image-to-image visual assistant from three key aspects: task definition, data construction, and model architecture. Our goal is to create a system that can precisely follow free-form user instructions for image generation, manipulation, and translation. Our PixWizard eliminates the need for task-specific design choices and achieves highly competitive performance across a diverse set of tasks, with strong generalization capabilities.

However, this work has some limitations. First, the current model architecture does not yet support multi-image input conditions, which is an increasingly important research area. Second, there is room for improvement in challenging tasks like segmentation and image grounding compared to specialized models. Additionally, the performance of the text encoder and foundation model also plays a crucial role. Better text encoding improves the model's ability to understand and execute instructions, while larger, more robust architectures enhance output quality. Notably, the modules and strategies proposed in PixWizard can be easily applied to other powerful text-to-image generators. In the future, we plan to explore more advanced diffusion models, such as SD3 and FLUX, and continue advancing this field toward a "GPT-4 moment".

## 6 ETHICS IMPACT

Modern text-to-image generation models, such as Stable Diffusion (Rombach et al., 2022b) and Lumina-T2I (Zhuo et al., 2024), raise ethical concerns, including social bias. PixWizard, which uses similar large-scale datasets, faces these same challenges, such as creating and spreading manipulated data or misinformation. Future work will focus on developing a responsible use framework, balancing the benefits of research transparency with the risks of open access, to ensure safe and beneficial use.

Notably, all datasets and benchmarks used in PixWizard were legally sourced from public repositories under academic-use licenses. Additionally, any extra image sources used in the paper have been responsibly sourced to ensure they are free, copyright-free, and commercially usable, preventing legal violations. We take full responsibility for adhering to data licenses and managing any legal risks.

## 7 ACKNOWLEDGEMENTS

We thank Ziheng Wu, Huan Teng, and Tonton Su for their valuable comments and suggestion to this project, including data construction, model discussions, and evaluation work. This project is funded in part by National Key R&D Program of China Project 2022ZD0161100, by the Centre for Perceptual and Interactive Intelligence (CPII) Ltd under the Innovation and Technology Commission (ITC)'s InnoHK, by NSFC-RGC Project N_CUHK498/24. Hongsheng Li is a PI of CPII under the InnoHK.

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

APPENDIX

## A  RELATED WORK

**Diffusion Models.** Diffusion models estimate the data distribution by modeling the gradient of the noise-perturbed data distributions (Sohl-Dickstein et al., 2015; Song & Ermon, 2019; Ho et al., 2020; Dhariwal & Nichol, 2021; Song et al., 2021). They have demonstrated remarkable performance in various fields, ranging from text-to-image generation (Rombach et al., 2022a; Saharia et al., 2022b; Betker et al., 2023; Jiang et al., 2024a; Zhang et al., 2023b), controllable generation (Zhang et al., 2023a; Ye et al., 2023), and image editing (Avrahami et al., 2022; Brooks et al., 2023; Kawar et al., 2023) to video (Ho et al., 2022; Brooks et al., 2024), audio (Kong et al., 2021; Huang et al., 2023), 3D (Guo et al., 2023; 2024), and motion (Tevet et al., 2023; Zhang et al., 2024b) generation. Beyond generation, recent works have also exhibited diffusion models' capabilities in computer vision tasks, such as semantic segmentation (Baranchuk et al., 2022; Xu et al., 2023), depth estimation (Ke et al., 2024b; Lee et al., 2024), image restoration (Xia et al., 2023), and reasoning enhancement Guo et al. (2025). Benefiting from the visual knowledge learned from large-scale pretraining, these works open up the potential for adapting pretrained diffusion models to downstream tasks in a generative manner, with excellent capabilities like handling inherent uncertainties and zero-shot generalization.

**Vision Generalists.** Building a vision generalist capable of unifying various visual tasks has been a long-standing goal. Inspired by the success of scaling sequential modeling with transformers in natural language processing, many vision generalists (Wang et al., 2022a; Chen et al., 2022b; Lu et al., 2022b; 2024; Mizrahi et al., 2024; Bachmann et al., 2024; Bai et al., 2024) follow a similar approach by converting inputs and outputs into sequences of discrete tokens (Van Den Oord et al., 2017; Esser et al., 2021), allowing joint modeling of different modalities within a unified framework. With the rapid advancements in large language models (LLMs) (Achiam et al., 2023; Touvron et al., 2023; Zhang et al., 2024c) and multi-modal large language models (MLLMs) (Gao et al., 2024; Zhang et al., 2024e;d; Jiang et al., 2024b; 2025), several works (Koh et al., 2024; Dong et al., 2024; Sun et al., 2023; Wu et al.; Han et al., 2023) have introduced task-specific tokens, aligning them with the embedding space of LLMs to equip text-only LLMs with the ability to perceive and generate images. However, a common drawback of these approaches is their limited performance on generation tasks, such as text-to-image generation and image editing. Additionally, their sampling efficiency is constrained by the next-token prediction paradigm, which worsens for image-to-image tasks or when working with high-resolution images. In contrast, diffusion models, known for their state-of-the-art generation performance and efficiency, are ideal candidates for image generation and manipulation tasks. Recent efforts (Wang et al., 2023c; Gan et al., 2024; Hu et al., 2024; Geng et al., 2024; Lei et al., 2025) aim to create a visual generalist by unifying multiple visual tasks through a natural language interface based on pretrained text-to-image diffusion. However, these models generally focus on a limited set of tasks within narrow domains and are constrained by the scalability limitations of early foundation models (Rombach et al., 2022b), limiting their potential as practical visual assistants. To address these challenges, we propose a solution from both the model and data perspectives to develop a more versatile visual assistant.

## B  MORE DETAILS FOR THE OMNI PEXEL-TO-PEXEL INSTRUCTION-TUNING DATASET

### B.1  IMAGE RESTORATION

All the open-source datasets used for the image restoration task are listed below.

| | | | | |
|---|---|---|---|---|
| ■ BSD (2001) | ■ RealBlur (2020) | ■ DPDD (2020) | ■ GoPro (2017) | ■ REDS (2019) |
| ■ DenseHaze (2019) | ■ NH-HAZE (2020) | ■ Reside-6K (2018) | ■ UHDM (2022) | ■ SIDD (2018) |
| ■ Rain1400 (2017) | ■ Outdoor-Rain (2019b) | ■ SSID (2022) | ■ RainDrop (2018) | ■ RainDS (2021) |
| ■ SRD (2017) | ■ RealSnow (2023) | ■ Snow100K (2018) | ■ CLWD (2021) | ■ CelebA-HQ (2015) |
| ■ RealSR (2019) | ■ LOL-v2 (2021) | ■ DIV2K (2017) | ■ FFHQ (2019) | ■ Flickr2K (2023a) |

### B.2 IMAGE GROUNDING

**(1) Segmentation Referring.** We define referring segmentation as highlighting the target object specified by the user in the output image. For example, if the model is given instructions like, "Please mark the pixels in {color} based on the referring description: {caption}," the resulting image would display a mask in the specified color over the appropriate object described in the caption. When constructing the data, we pre-set the mask to a specific color with 50% opacity and apply it directly to the original image. This method makes it easier for humans to evaluate the accuracy of the predicted mask.

**(2) Box Detection Referring.** Instead of pixel-level grounding, we use bounding boxes to highlight the target object specified by the user in the output image. Prompts for this task include instructions like, "Mark the specified area with a bounding box in {color}: {caption}." The model then frames the described object with a bounding box in the specified color. Similar to referring segmentation, we pre-set the bounding box to a specific color and apply it directly to the original image to produce the final output. During inference, we follow the post-processing methods outlined in InstructCV (Sec. A.3) (Gan et al., 2024) to derive the coordinates of the specified region from the output image.

**(3) Binary Mask Prediction.** To promote the use of referring segmentation in real-world scenarios, we shift the objective from rendering images to directly predicting a binary mask. The prompt for this task might be: "Generate a binary mask for the described object: {caption}." The model is expected to produce a binary mask image where the object described in the caption is represented as a white region, with the background in black, excluding any original image content. Since the binary mask is a single-channel image, we replicate the mask across three channels to convert it into RGB space during training.

### B.3 CONTROLLABLE GENERATION

**Canny Edge to Image.** We use a Canny edge detector (Canny, 1986) (with random thresholds) and a Tiny and Efficient Edge Detector (TEED) (Soria et al., 2023) to obtain 1M canny-image-caption pairs from our collected natural images.

**Holistically-Nested Edge(HED) Boundary to Image.** We use HED boundary detection (Xie & Tu, 2015) to obtain 1M edge-image-caption pairs from our collected natural images (a part of images are source of the Canny Edge dataset.)

**Depth Map to Image.** Depth information is crucial for producing images with a sense of three-dimensionality. We used the Depth Anything V2 model (Yang et al., 2024b) obtain 1M depth-image-caption pairs, enabling accurate generation of depth maps across different visual scenarios.

**User Sketch to Image.** Following ControlNet (Zhang et al., 2023a), we generate human sketches from images by applying HED boundary detection (Xie & Tu, 2015) combined with strong data augmentations, including random thresholds, random masking of sketch portions, random morphological transformations, and random non-maximum suppression. This process results in 0.4 million sketch-image-caption pairs.

**Human Pose to Image.** For human pose-based generation, we employed the OpenPose model (Cao et al., 2017) for real-time multi-person 2D pose estimation. To ensure quality, only images where at least 30% of the key points of the whole body were detected were retained; those with fewer detected key points were discarded. We directly use visualized pose images with human skeletons as input condition. Finally, we obtain 0.25M pose-image-caption pairs.

**Semantic Segmentation to Image.** The semantic mask annotation is produced using OneFormer (Jain et al., 2023), as adopted in ControlNet-1.1 [1], providing precise segmentation maps as conditions for image generation. This process results in 1M seg-image-caption pairs.

**Normal Map to Image.** The surface normals are generated using DSINE (Bae & Davison, 2024), which contributed to the depiction of surface orientation and texture details in the generated images. Finally, we obtain 0.8M normal-image-caption pairs.

---

[1] https://github.com/lllyasviel/ControlNet-v1-1-nightly

**Line-art to Image.** We use a cartoon line drawing extraction method (Xiang et al., 2021) to generate line drawings from cartoons. This process yields 0.8M normal-image-caption pairs.

### B.4 DENSE IMAGE PREDICTION

**Depth Estimation.** Monocular depth estimation is a dense prediction task to estimate the per-pixel depth value(distance relative to the camera) given an input RGB image. The datasets we use include Hypersim (Roberts et al., 2021) and VKITTI 2 (Cabon et al., 2020), and our collected depth maps from controllable dataset. For different datasets, the ground-truth depth for each pixel may vary, being either an absolute depth value or a relative depth value. Here we uniformly map the ground-truth value from real-valued range to the integer RGB space with range $[0, 255]$, and let the three channels be the same ground truth. During inference, we directly average the outputs of the three channels, and then perform an inverse linear transformation specific to each dataset to obtain a depth estimate in the real-valued range.

**Surface Normal Estimation.** For surface normal estimation, we train the model to generate RGB images that encode pixel orientations differently. We convert the $x/y/z$ orientations into $r/g/b$ values to create a normal map visualization. The datasets we use include NYUv2 (Silberman et al., 2012) and our collected surface normals maps from controllable dataset.

**Semantic Segmentation.** Semantic segmentation is a dense prediction task to predict the per-pixel semantic label given an input image. Given a semantic segmentation task, we formulate different semantic categories using different colors in RGB space. Specifically, we define the background and ignore areas as black, i.e., pixels in color $(0, 0, 0)$, and generate the colors for foreground categories using the pre-defined color-label dictionary. During inference, to decode the output image to a single channel map where each pixel represents a class label, we compute the L2 distance between each output pixel and the pre-defined colors for each semantic category, and take the label of the closest color as the predicted category. The datasets we use is ADE20K (Zhou et al., 2017b), which covers a broad range of 150 semantic categories.

### B.5 INSTRUCTION PROMPT EXAMPLES.

Handling free-form user prompts, rather than relying on fixed task-specific prompts, significantly enhances PixWizard's flexibility and usability as a general visual assistant. In Fig. 10, we showcase various examples where task-specific templates are dynamically rephrased and adapted by GPT-4o

## C MORE DETAILS FOR THE ARCHITECTURE

### C.1 PRELIMINARIES

**Diffusion Transformers.** Diffusion models (Sohl-Dickstein et al., 2015; Ho et al., 2020; Dhariwal & Nichol, 2021; Song et al., 2021) are a family of generative models demonstrating remarkable performance in modeling data distributions. These models are trained to estimate clean data samples (or added noise) from their noisy version perturbed by a pre-defined Gaussian noise schedule. During sampling, they can generate data samples by iterative denoising starting from prior Gaussian distributions. Recent advancements in diffusion models, such as SoRA (Brooks et al., 2024), SD3 (Esser et al., 2024), and PixArt (Chen et al., 2024b;a), exhibiting a paradigm shift from the classic diffusion U-Net architecture to diffusion transformers (DiTs) (Peebles & Xie, 2023). DiTs appear to be a unified architecture with minimal modifications to original transformers (Vaswani et al., 2017). Therefore, these models demonstrate better scaling properties and can be naturally extended to more modalities, such as integrating text and image conditions through cross-attention.

**Flow-based Models.** Another line of generative models that extends the definition of diffusion models is ODE-based continuous normalizing flows (Chen et al., 2018), which are also known as flow matching (Lipman et al., 2023; Tong et al., 2024; Albergo & Vanden-Eijnden, 2023) or rectified flows (Liu et al., 2023b). Specifically, given the data space $\mathbb{R}^d$ with samples $x \in \mathbb{R}^d$, flow-based models aim to learn a time-dependent velocity field $v : [0, 1] \times \mathbb{R}^d \to \mathbb{R}^d$ leading to a flow $\phi : [0, 1] \times \mathbb{R}^d \to \mathbb{R}^d$ that can push noise $x_0 \sim p_0(x_0)$ from source distribution to data $x_1 \sim p_1(x_1)$ from target distribution. This velocity field and associated flow can be defined by an ordinary

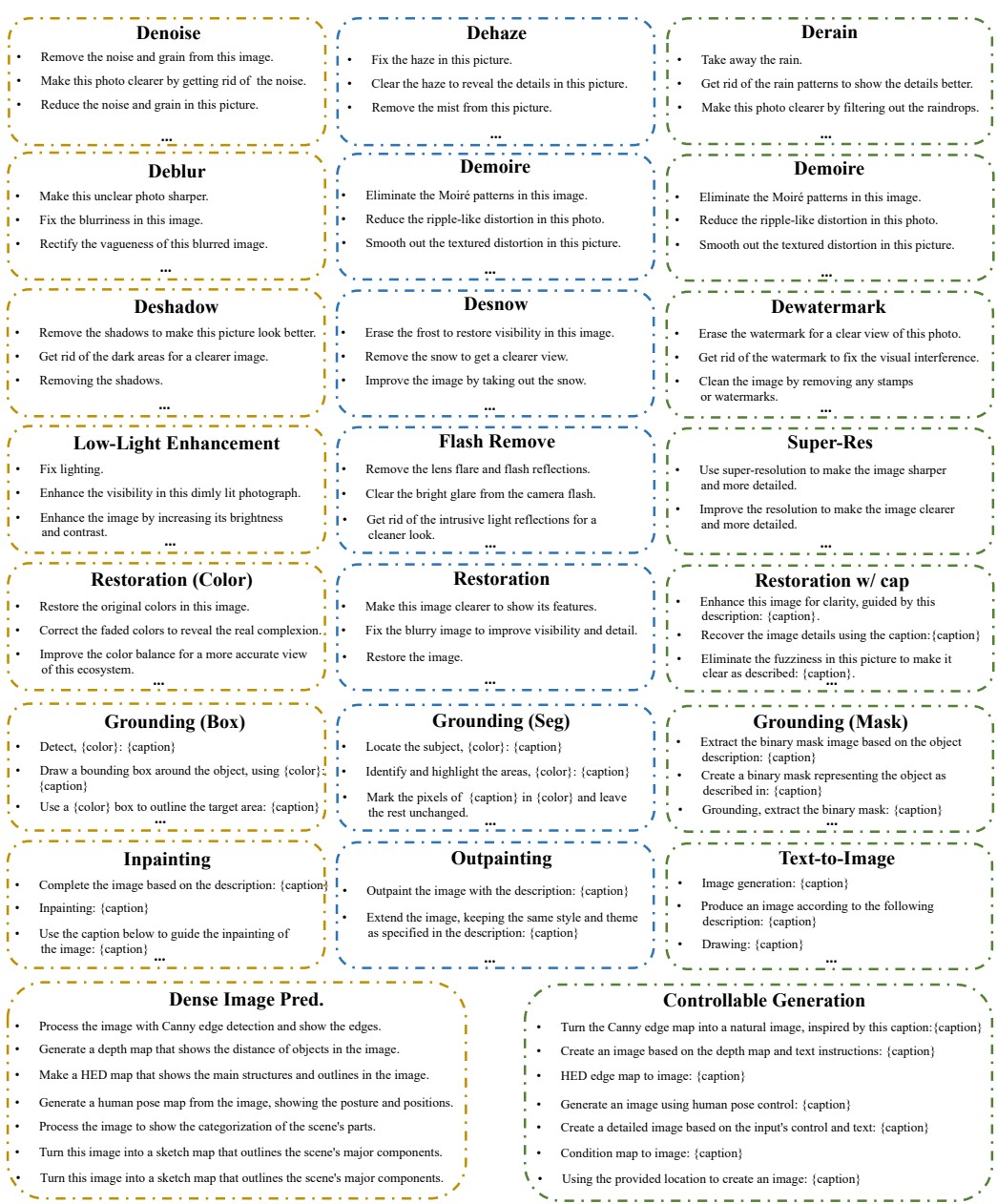

Figure 10: Examples of GPT4o-paraphrase user prompts for different task.

differential equation (ODE): $\frac{d}{dt}\phi_t(x) = v_t(\phi_t(x), t)$ where $\phi_0(x) = x$. Similar to the denoising network in diffusion models, the Flow Matching (FM) objective trains a time-dependent network $v_\theta(x_t, t)$ to regress against the ground truth velocity field $u_t(x_t, t)$. However, direct computation of this FM objective is intractable in practice, since there is no closed-form solution of $u_t(x_t, t)$. Instead, we can minimize the tractable Conditional Flow Matching (CFM) objective defined as:

$$\mathcal{L}_{\text{CFM}}(\theta) = \mathbb{E}_{t, p_1(x_1), p_t(x_t|x_1)} \|v_\theta(x_t, t) - u_t(x_t, t|x_1)\|^2, \tag{7}$$

where $t \sim \mathcal{U}(0, 1)$, $x_1 \sim p_1(x_1)$, and $x_t \sim p_t(x_t|x_1)$. It has been validated that the FM and CFM objectives share identical gradients with respect to $\theta$, while CFM offers the flexibility to choose the design choices of $u_t(x_t|x_1)$ and $p_t(x_t|x_1)$. A natural choice is to build the conditional probability paths as straight paths between the source and target distributions (Liu et al., 2023b; Lipman et al.,

2023), *i.e.*, $x_t = tx_0 + (1-t)x_1$. We can then use Equation 7 for training and solve the ODE from $t = 0$ to $t = 1$ to sample new data points.

## C.2 TASK EMBEDDER

As shown in Fig. 11, introducing the task embedder helps adaptively cluster similar task instructions in the latent space while separating those from different tasks, guiding the model's generation process in the correct direction.

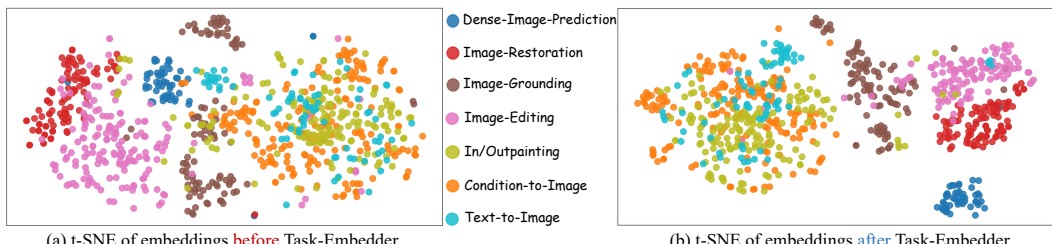

(a) t-SNE of embeddings before Task-Embedder    (b) t-SNE of embeddings after Task-Embedder

Figure 11: t-SNE visualization of the global text embeddings. Each dot represents a user instruction.

## C.3 CLASSIFIER-FREE GUIDANCE FOR MULTI-MODAL CONDITIONS

Classifier-free guidance (Ho & Salimans, 2022) is a technique that balances quality and diversity in images generated by a diffusion model. For our PixWizard, the network $e_\theta(z_t, c_I, c_T)$ has two conditionings: the input image $c_I$ and text instruction $c_T$. Similar to InstructPix2Pix (Brooks et al., 2023) and InstructCV (Gan et al., 2024), we employ a tailored noise predictor that assigns different weights, $w_I$ and $w_T$, to different conditionings, which can be adjusted to trade off how strongly the generated samples correspond with the input image and how strongly they correspond with the edit instruction. During training, we randomly set $c_I = \varnothing_I$ or $c_T = \varnothing_T$ for 5% of examples, and both conditions are $\varnothing$ for 5% of examples. The process is as follows:

$$
\begin{aligned}
\tilde{e}_\theta(z_t, c_I, c_T) = \ &e_\theta(z_t, \varnothing, \varnothing) \\
&+ w_I \cdot (e_\theta(z_t, c_I, \varnothing) - e_\theta(z_t, \varnothing, \varnothing)) \\
&+ w_T \cdot (e_\theta(z_t, c_I, c_T) - e_\theta(z_t, c_I, \varnothing))
\end{aligned}
\tag{8}
$$

In Fig. 12, we illustrate the effects of the two parameters on the generated samples. As shown, changes in both $W_I$ and $W_T$ significantly influence the results. Specifically, in PixWizard, $W_I$ strongly affects the color distribution of the generated images. An increase in $W_I$ leads to more saturated and unrealistic colors. This can also negatively impact tasks such as depth estimation, where subtle depth variations may be represented as solid black regions. In contrast, the effects of $W_T$ vary depending on the task. For example, in dense image prediction tasks, increasing $W_T$ can degrade the quality of the estimation. However, for generative tasks, a higher $W_T$ typically has little effect.

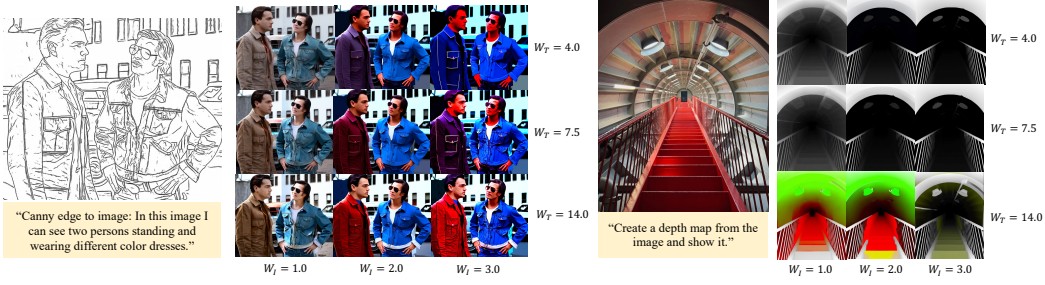

Figure 12: Impact of classifier-free guidance.

### C.4 Any Resolution

PixWizard inherits the dynamic partitioning and padding scheme introduced by (Zhuo et al., 2024), allowing the model to handle images of any resolution and aspect ratio during both fine-tuning and inference.

The dynamic partitioning and padding scheme is introduced to optimize image token handling, avoiding fixed resolutions. Specifically, given constraints on the maximum number of patches and the maximum aspect ratio, the method defines a set of candidate patch partitions. For each input image size, it determines the optimal patch partition based on a matching ratio between the input size and the candidate partition, selecting the best fit. The input image is then resized accordingly using the chosen partition. This dynamic process leads to varying sequence lengths of patch tokens, so padding is applied to align the lengths across batches using a pad token. To prevent unwanted interactions between pad and regular tokens, attention masks are also introduced. This approach maximizes efficiency while preserving image integrity during embedding.

However, in practice, the required resolutions for different tasks can vary significantly. To support more flexible handling of arbitrary resolutions while preserving the original resolution as much as possible, we use $[512^2, 768^2, 1024^2]$ as resolution centers to generate candidate patch partitions. During training, we group data with similar resolutions into buckets, ensuring that sequence lengths within each batch are comparable, minimizing padding tokens, and improving training efficiency. During inference, by incorporating NTK-Aware Scaled RoPE (Peng & Quesnelle, 2023) and sandwich normalization, PixWizard demonstrates exceptional resolution extrapolation.

## D  More Experimental Results

### D.1  Implementation Details

**Dense Image Prediction.**  For semantic segmentation, we assign labels by identifying the nearest neighbor RGB color value, and accuracy is evaluated using the Mean Intersection over Union (mIoU) metric. For monocular depth estimation, we average the output image across the three channels and apply the inverse of the linear transformation used during training to obtain depth estimates within the range of $[0, 10]$ meters. Accuracy is evaluated using the Root Mean Square Error (RMSE). For surface normal estimation, we recover the corresponding normal vectors from the output image and use the Mean Angle Error to assess accuracy.

**Controllable Generation.**  We mainly evaluated PixWizard's ability based on two conditions: canny edge maps and depth maps. Following ControlNet++ (Li et al., 2024), we measured controllability using RMSE for depth maps and F1-Score for canny edges, comparing input conditions with features from the generated images. Image quality and text alignment were assessed using FID (Fréchet Inception Distance) and CLIP-Score. All experiments were conducted at a resolution of $512 \times 512$.

**Image Inpainting.**  We used latent diffusion (Rombach et al., 2022b) to measure FID and LPIPS, focusing on samples where 40-50% of the image area was inpainted. For outpainting, we followed MaskGIT (Chang et al., 2022a) settings, extending the image by 50% and evaluating performance with FID and Inception Score (IS) on $512 \times 512$ crops from the Places dataset (2017a).

**Text-to-image Generation.**  We evaluated PixWizard with two methods: visual examples and automatic metrics, including the Human Preference Score (HPS) v2 (Wu et al., 2023) and Zero-shot FID-30K on the MS-COCO dataset (Lin et al., 2014).

### D.2  Image Restoration

Following previous works (Conde et al., 2024; Potlapalli et al., 2024; Ai et al., 2024), besides the results from the deraining and denoising benchmarks, we selected six additional image restoration tasks to further evaluate the robustness of PixWizard: Snow100K-L (Liu et al., 2018) for desnowing, Reside (outdoor) SOTS (Li et al., 2018) for dehazing, LOLv2 (Yang et al., 2021) for low-light enhancement, and GoPro (Nah et al., 2017) for deblurring. Additionally, to assess zero-shot capabilities on tasks not encountered during training, we use TOLED (Zhou et al., 2021) for under-display camera (UDC) image restoration and UIEB (Li et al., 2019a) for underwater (UW) image restoration. We evaluate performance using PSNR and SSIM as distortion metrics.

Table 5: Quantitative results on 6 restoration tasks with existing image restoration methods.

| Methods | Desnowing Snow100K-L PSNR↑ | SSIM↑ | Dehazing SOTS PSNR↑ | SSIM↑ | Low-light Enh. LOLv2 PSNR↑ | SSIM↑ | Deblurring GoPro PSNR↑ | SSIM↑ | [Zero-shot](UDC)IR. TOLED PSNR↑ | SSIM↑ | [Zero-shot](UW)IR. UIEB PSNR↑ | SSIM↑ |
|---|---|---|---|---|---|---|---|---|---|---|---|---|
| SwinIR (2021) | - | - | 21.50 | 0.891 | - | - | 24.52 | 0.773 | - | - | - | - |
| Restormer (2022) | 30.98 | 0.914 | 24.09 | 0.927 | 20.77 | **0.851** | 27.22 | 0.829 | 27.74 | 0.841 | **17.34** | **0.770** |
| NAFNet (2022a) | **31.42** | **0.920** | 25.23 | 0.939 | 18.04 | 0.827 | 26.53 | 0.808 | **27.90** | **0.848** | 17.31 | 0.736 |
| AirNet (2022) | 30.14 | 0.907 | 21.04 | 0.884 | 19.69 | 0.821 | 24.35 | 0.781 | 26.76 | 0.799 | 17.09 | 0.761 |
| PromptIR (2024) | 30.91 | 0.913 | **30.58** | **0.974** | 21.23 | 0.860 | 27.02 | 0.798 | - | - | - | - |
| DA-CLIP (2023) | 28.31 | 0.862 | 30.16 | 0.936 | **21.76** | 0.762 | 24.65 | 0.703 | - | - | - | - |
| InstructIR (2024) | - | - | 26.90 | 0.952 | - | - | **29.70** | **0.892** | - | - | - | - |
| PixWizard | 29.66 | 0.883 | 28.14 | 0.937 | 20.29 | 0.807 | 24.68 | 0.774 | 27.22 | 0.826 | 16.99 | 0.752 |

**Results.** Table 5 presents a additional performance comparison with state-of-the-art (SOTA) task-specific and all-in-one restoration methods. As shown in the results, even though image restoration data constitutes only a small portion of the overall training data, our method still demonstrates competitive performance, even outperforming some task-specific methods. For example, it outperforms DA-CLIP in the desnowing task and exceeds NAFNet, Restormer, and AirNet in the dehazing task. Furthermore, when evaluated on tasks not included in the training phase, our method achieved performance comparable to that of specialized models, highlighting the generalizability of our approach.

## D.3 IMAGE GROUNDING

We evaluate referring segmentation tasks on the gRefCOCO (gRef) (Liu et al., 2023a), RefCOCO, and RefCOCO+ validation and test sets. To assess the performance gap between our approach and specialized models, we report results from several expert methods and primarily compare our approach with two unified models: Unified-IO (Lu et al., 2022b) and InstructDiffusion (Geng et al., 2024). Unified-IO directly produces the corresponding binary mask, while InstructDiffusion requires a post-processing network to extract masks from the output image. We use two comparison methods: ($i$) We convert the original image to the HSV color space to enhance the mask's hue, then apply a threshold for extraction. These results are reported as w/ HSV. ($ii$) We directly generate the corresponding binary mask for comparison. Following standard practices (Liu et al., 2023a), we use cumulative IoU (cIoU) to measure performance.

Table 6: Quantitative results on referring segmentation in terms of cIoU.

| Method | gRefCOCO val | RefCOCO val | test A | test B | RefCOCO+ val | test A | test B |
|---|---|---|---|---|---|---|---|
| CRIS 2022c | 55.34 | 70.47 | 73.18 | 66.10 | 62.27 | 68.08 | 53.68 |
| LAVT (2022) | 57.64 | 72.73 | 75.82 | 68.79 | 56.86 | 62.29 | 48.14 |
| ReLA (2023a) | 62.42 | 73.21 | 75.24 | 68.72 | 56.10 | 62.26 | 47.89 |
| Unified-IO (2022a) | 17.31 | 46.42 | **46.06** | **48.05** | **40.50** | **42.17** | 40.15 |
| InstructDiffusion w/ HSV (2024) | 33.19 | 41.64 | 40.81 | 41.98 | 33.20 | 37.85 | 26.92 |
| PixWizard w/ HSV | **33.65** | **47.28** | 44.12 | 45.38 | 40.07 | 38.89 | **40.76** |
| PixWizard | 29.07 | 44.38 | 40.13 | 42.09 | 36.49 | 35.30 | 38.93 |

**Results.** Table 6 reports the results for referring segmentation. Our model demonstrates strong performance under two different mask extraction methods, outperforming InstructDiffusion across almost all evaluation datasets. In the various test sets of RefCOCO and RefCOCO+, our performance is comparable to Unified-IO, but on gRefCOCO, we significantly outperform Unified-IO (33.65 vs. 17.31). However, it is important to note that these unified methods still lag considerably behind those specifically designed for referring segmentation. Additionally, we observed that using a generative model for grounding tasks presents a significant challenge due to the model's limited perception and localization capabilities. This often results in failures to correctly locate objects when multiple targets are present, highlighting the need for improvements in instruction and understanding in future work.

### D.4 MORE PRECISE EDITING

Based on the robust inpainting and grounding capability of PixWizard, we find that it allows user for more precise image editing tasks: *(i) Remove Anything.* It tackles the object removal problem (Criminisi et al., 2003; 2004; Elharrouss et al., 2020), enabling users to seamlessly remove specific objects. As shown in Fig. 13, PixWizard first generates a target mask based on user instructions, then fills the area with appropriate background details. *(ii) Replace Anything.* It lets users swap objects in an image, replacing them with specified objects while maintaining background consistency. *(iii) Add Anything.* This allows users to insert objects into an image. By adding a mask and providing a text prompt, PixWizard generates the desired content using its advanced inpainting ability.

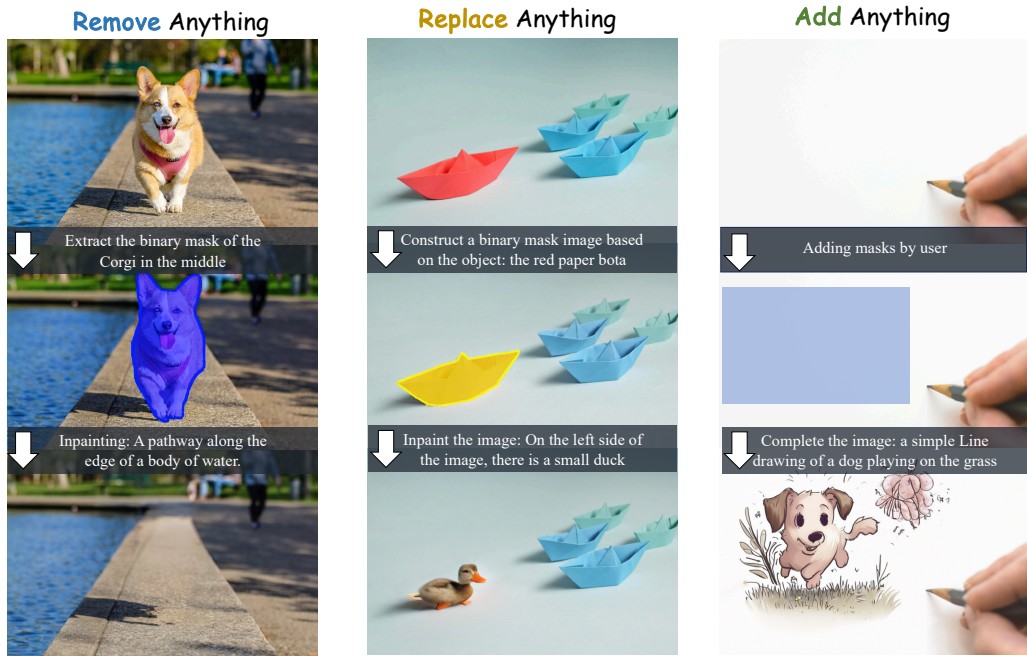

Figure 13: Visualization results of **Remove**, **Replace** and **Add** Anything.

### D.5 LIMITATIONS OF IMAGE EDITING TASKS

Image editing tasks can generally be divided into rigid and non-rigid edits. Rigid editing typically involves simpler modifications such as adding, removing, replacing, style transferring, or altering the texture of an object while maintaining the overall spatial structure of the image. On the other hand, non-rigid editing involves more complex transformations that require spatial manipulation, such as altering the pose of an object, swap the position or changing the viewpoint.

Despite the versatility of PixWizard in handling multiple types of rigid editing, we found that PixWizard currently exhibits limitations when dealing with non-rigid editing tasks, as several visualization examples shown in Figure. 14. When users attempt to provide more complex editing instructions, PixWizard often fails to execute them successfully, instead keeping the original image largely unchanged. This limitation arises primarily due to two reasons: the design of the structural-aware guidance module and the lack of diverse training data for non-rigid transformations. PixWizard's structural-aware guidance is implemented by concatenating the VAE latent representation with random noise along the channel dimension. We found that this method would limit generation directions to follow the structure of the input image, constraining its ability to perform non-rigid transformations that demand substantial structural alterations. However, it is worth noting that the "concat" method is efficient for other visual tasks such as image estimation, low-level tasks, and image grounding, which require maintaining similar structural characteristics. This is one of the reasons we initially adopted it. Furthermore, the datasets used to train PixWizard—such as UltraEdit (2024), MagicBrush (2024a), HQ-Edit (2024), Instruct P2P (2023), and SEED-X-Edit (2024)—focus predominantly on rigid editing tasks and lack non-rigid editing data pairs.

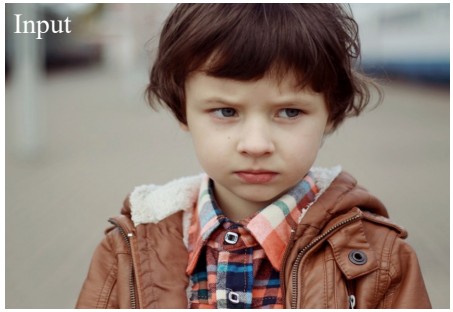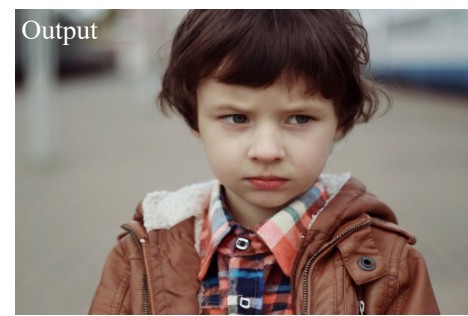

Make the boy close his eyes and smile.

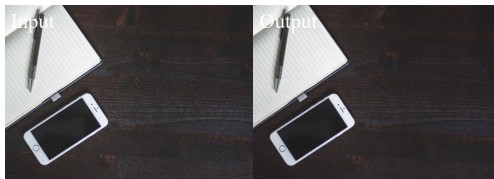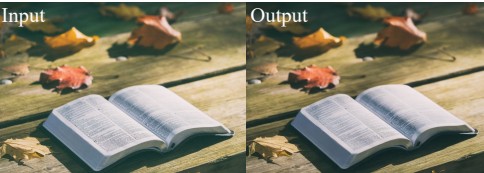

Move the iPhone in the image to the right side of the image.          Please help me close the book.

Figure 14: Bad cases of non-rigid editing.

How to perform non-rigid editing in PixWizard is indeed one of the directions we aim to address. Here are some of our current thoughts: (1) Constructing non-rigid editing data pairs is important. This could involve utilizing state-of-the-art editing methods like Imagic (Kawar et al., 2023) and Forgedit (Zhang et al., 2023c), or advanced inpainting methods, complemented by GPT-assist and manual filtering to create high-quality non-rigid editing pairs. (2) Refining the structural-aware guidance. The primary objective is to move away from directly concatenating the VAE latent in the channel dimension. In future work, we aim to explore the integration of flow-based image inversion techniques and shift towards fusing targeted image features within the self-attention layers of the DiT block. Further experiments will be conducted to validate the effectiveness of this approach.

It is also worth mentioning that we found that the quality of most existing open-source image editing datasets is subpar. There are two main issues: first, the target images often contain artifacts, making them look unrealistic; second, there is a lack of alignment between the target images and the editing instructions. Through experiments, we discovered that simply using more open-source data does not necessarily lead to better performance. On the contrary, using a small amount of high-quality editing data results in better performance in both evaluation metrics and user experience with PixWizard. Therefore, it is both crucial and meaningful to explore methods for effectively cleaning open-source editing datasets. We are committed to addressing this issue in the future.

## D.6 GENERALIZATION ABILITY TO UNSEEN TASKS AND PROMPTS

**Zero-shot Examples.**    We have provided several zero-shot visual examples in Fig. 15. These include marking target objects using different symbols other than a box, such as a star, a point, or a circle. Additionally, we can change the color of detection boxes to colors that were not seen during training. Furthcerermore, we can simultaneously perform multiple tasks, such as joint deraining and watermark removal, deraining and super-resolution, or inpainting and outpainting, with masks of any shape and mixed black-and-white colors. These examples highlight the generalizability of our approach. However, getting PixWizard to perform entirely new tasks that are completely unrelated to the training data, such as adding an object and then detecting it or contour detection, is challenging. This limitation points to the boundaries of PixWizard's generalization capabilities.

**Few-shot Examples.**  Following Emu Edit's experimental setup (Sheynin et al., 2024), we further validated PixWizard's generalizability from a few-shot learning perspective. We prepared an object contour detection task as an illustration. We constructed 50 training samples for this task (all with red contours) and added them to our training dataset, then fine-tuned PixWizard. As shown in Fig. 15, PixWizard quickly acquired this capability. Additionally, we changed the contour color and found that PixWizard could still follow the instructions, implying that the model can effectively generalize to novel tasks.

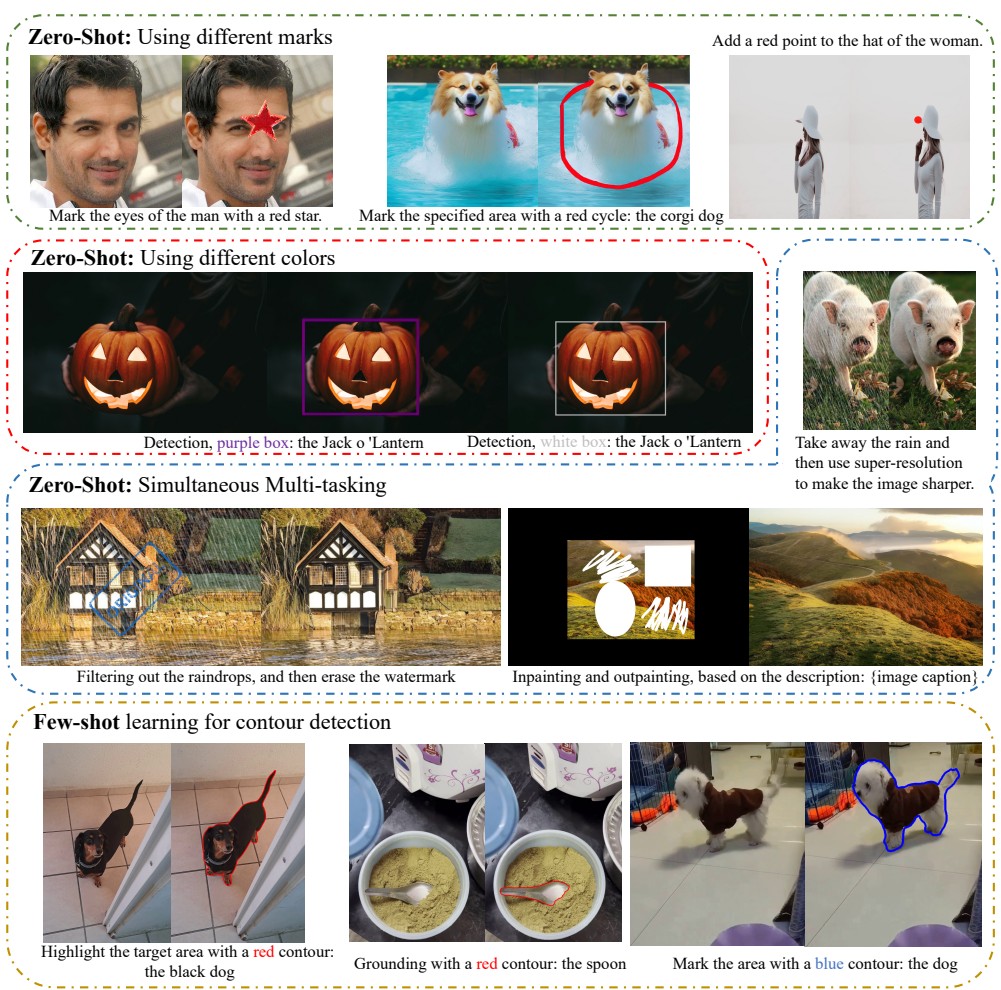

Figure 15: Visualization results of zero-shot and few-shot.

## D.7 MULTI-HOT GUMBEL-SOFTMAX

In the implementation of the Multi-Hot Gumbel-Softmax (MHGS), the pseudocode is defined as follows:

```python
def MHGS(logits, temp=1, dim=-1, sample_tokens=16):
    # Add Gumbel noise and scale by temperature
    gumbels = (logits + GumbelNoise(shape=logits.shape)) / temp

    # Apply Softmax to obtain soft outputs
    y_soft = Softmax(gumbels, dimension=dim)

    # Select top-k values for discrete output
    indices = Top-K(y_soft, k=sample_tokens, dimension=dim)

    # Create a hard multi-hot tensor from indices
    y_hard = MultiHotTensor(indices, shape=logits.shape)

    # Combine hard and soft outputs while preserving gradients
    ret = y_hard - StopGradient(y_soft) + y_soft

    return ret
```

