# OpenReview forum: "PixWizard: Versatile Image-to-Image Visual Assistant with Open-Language Instructions"
_ICLR.cc/2025/Conference — ICLR 2025 Poster_

### Official Review · Reviewer_kiLh · 2024-10-17

**Soundness:** 3
**Presentation:** 3
**Contribution:** 2
**Rating:** 6
**Confidence:** 4

**Summary:**

The authors propose PixWizard, a flow-based model that can tackle a range of visual tasks including text-to-image generation, inpainting and outpainting, as well as a long list of image-to-image translation tasks (sketch-to-image, depth-to-image, denoising, box-drawing based detection, segmentation and more).

The core of their approach revolves around teaching a pre-trained text-to-image model (Lumina-Next) to tackle multiple novel tasks by conditioning it on additional image input features and features describing the specific tasks the network should handle. The authors propose a set of blocks to extract said features and integrate them into the network, and propose a data curation and balancing strategy to ensure the model does not ignore rare tasks.

Finally, the authors conduct a wide range of experiments on many different tasks and demonstrate that their approach is competitive (and sometimes even improves over) networks trained for each individual task, and also outperforms most generalist network approaches which aim to concurrently tackle many tasks.

**Strengths:**

The paper does a remarkable job in evaluating against a long range of baselines and across a wide list of tasks. It also demonstrates good performance across most of the tasks, showing that their approach can indeed lead to a fairly general multi-task model.

The authors also provide a fairly detailed overview of their network, block choices, and the sources and handling of their training data. In the appendix, they further analyze many of their contributions in an ablation study, and report on choices even when they do not meaningfully impact the results (which is good in my eyes, as it lets future work know which components they may want to discard in favor of simplicity).

Finally, the appendix also contains an interesting application for the all-in-one approach, where they use the model to segment parts of the image and then edit them. Such interactive multi-step approaches that use different network skills to improve results can serve as a good reason for exploring multi-task models. Please see more on this in the weakness section below.

**Weaknesses:**

My chief concerns with the paper are as follows, ranked from most-important to least important:

1) The paper does not appear to distinguish itself from prior, published multi-task approaches like Emu Edit [Sheynin et al, CVPR 2024], and it does not actually outperform them. More specifically, Emu Edit already introduced a multi-task approach based on a pre-trained text-to-image model. It already has learned task embeddings, and it shows a range of similar downstream tasks.
I appreciate that outperforming closed models trained on unknown datasets is difficult, but when I ask myself what I learned from reading this paper that I did not already know from Emu Edit, I have a hard time coming up with an answer, and this is an issue.

2) The paper claims to generalize to unseen tasks (e.g. L26). I may have missed the results that corroborate this claim, but the closest I could find are the zero-shot restoration experiments in the appendix. However, these are close enough to existing restoration tasks that even non-multi-task baselines can generalize well to them and outperform this model? I would have liked to see generalization to actual new tasks, even using inverted task embeddings like Emu Edit. Otherwise, this weakens the contribution.

3) Related to (1) – most of the paper focuses on comparisons to other methods, to the point where ablation is relegated to the end of the appendix. I think ablation is crucial here and should be moved into the core paper, even at the cost of pushing out some experiments. since it is the only section that gives us validation for the specific new ideas introduced by the paper. This method uses a different base-model, is trained on different data, and has a significantly different number of parameters from the baselines. I understand that you beat them, but I also want to understand why.

4) There is relatively weak motivation for why multi-task models are needed. Saving some memory compared to specialized approaches is okay, but its not a significant contribution. Again, this can be contrasted with Emu Edit which show that training on multiple tasks can improve each of them individually, and that training on discriminative tasks like segmentation can also improve performance on generative tasks like editing. I had a hard time finding similar motivations here.

5) The paper can be better self-contained. Some components (e.g., dynamic partitioning and padding scheme) refer back to recent preprints with no citations, and if these parts are important then they should probably be given a bit more detail (even in the preliminaries in the appendix).


Minor issues that did not affect my score:

1) L121: without extra any -> without any extra

2) “Next, we concatenate the image latent with the noise latent along the channel dimension” – Consider citing SR3 [Saharia et al, TPAMI 2022] or Palette [Saharia et al, SIGGRAPH 2022].

3) Table 3 – Bold numbers are missing from the baselines when they win.

4) L483 notes that the method outperforms the inpainting baselines based on FID and LPIPS scores, but the method seems to underperform the baselines on LPIPS?

5) Appendix C.4. – It seems like the gains highlighted here are not related to any change in the proposed paper, but are just due to selecting a flow-matching baseline over a diffusion model? Did I miss something?

6) L1544: less tokens requiring less inference time should be shown experimentally because it’s not trivial that the gain here is meaningful.  The method uses extra layers to actually select what tokens to drop, and the tokens enter through cross-attention (as opposed to self-attention) which means they scale compute linearly and not quadratically?

**Questions:**

I tried to outline my concerns in the weakness section above. To sum them up in two main questions:

1) What does this paper bring to the table that doesn’t already exist in prior published work like Emu Edit?

2) What is the extent of generalization that your model actually supports? Can you back this up with experiments?

I appreciate the effort the authors clearly put into the work, so if these are answered in a satisfactory manner, I’ll gladly increase my score.

Minor clarification questions (skip these if you don’t have time):

3) What are the details of the MHGS? How is it implemented?

4) L100 says “We extend the dynamic partitioning and padding scheme to handle input images of any resolution, aligning closely with human perception.” but L279 says these resolution capabilities are inherited from [Zhuo et al., 2024]. Do you mean that Zhou support multiple resolutions but not good enough, and you improve this? Are you extending the scale of resolutions that they support?

5) L298: “we assign manual sampling weights to each dataset” How? Can you describe your criteria?

---

> ### Author Response · Authors · 2024-11-21
> **Response [1/3] to Official Review by Reviewer kiLh**
>
> > **Q1 and W1: What does this paper bring to the table that doesn’t already exist in prior published work like Emu Edit?**
>
> We think there are major differences between our PixWizard and Emu Edit:
>
> 1. We have expanded our focus to include a broader range of visual tasks, aiming to explore whether the image-to-image paradigm has the potential to train a generalist model with satisfying performance. In contrast, Emu Edit focus more on editing tasks. We have detailed how different visual tasks can be transformed into a unified input-output training format and how targeted postprocessing methods can convert them back to their original formats. Our experimental results demonstrate that generative models can effectively handle various visual tasks through appropriate format conversions. Furthermore, the stronger the pre-trained generative model, the better the overall performance after fine-tuning. For example, the overall experimental results of PixWizard surpass those of InstructPix2Pix and InstructDiffusion, both based on SD1.5 initialization. This highlights the potential for further exploration with more powerful models, such as FLUX, to develop even stronger generalist models.
>
> 2. Regarding how different tasks influence each other within a unified multi-task framework, our findings differ from Emu Edit. In the Emu Edit paper, the authors noted that "multi-task learning leads to enhanced performance in each individual task," but their experiments only validated the influence on image editing, leaving the impact on other tasks unknown. We made the following observations:
>   - A general model trained on multi-tasks does not always outperform the model trained on a single task. We found that some tasks, such as image editing and image grounding, show improved performance with multi-task learning, aligning with the experimental results of Emu Edit. However, some vision tasks do not see performance gains from multi-task learning. The example are dense image estimation tasks, as shown in the results below. One possible reason is that their output distributions differ significantly from other tasks, making multi-task learning less beneficial for them.
>   - The text-to-image task can effectively enhance the performance of inpainting, outpainting, and controllable image generation tasks. We believe that this is because these tasks require general image generation capability. Introducing text-to-image strengthens the model's generative ability. However, multi-tasking does not further improve the performance of the text-to-image task.
>
> | |Depth Est. (RMSE↓) |Sem. Seg (mIoU↑) |Derain (SSIM↑) |Grounding (cIoU↑) |Emu Edit (CLIPout↑) |Canny-to-image (FID ↓)|Inpainting (FID ↓)| T2I (HPSv2↑)|
> |--|--|--|--|--|--|--|--|--|
> | **Expert**    | **0.281** | **38.14**| **0.918** | 43.33 | 0.237 | 15.98 |10.32 | 27.47 |
> | **General**   | 0.287 | 36.76 | 0.917 | **46.44**  | **0.248** | **15.66** | **9.27** | 27.47 |
>
> 3. PixWizard is the first to address a common issue faced by all current instruction-based image-to-image models: the text-to-image capability is no longer supported after fine-tuning on image-to image data. In contrast, we unified the text-to-image task within the image-to-image paradigm by introducing a blank canvas and verified its feasibility and effectiveness. As a result, users can seamlessly utilize the pre-trained model's text-to-image generation capabilities without needing to switch models. Additionally, as highlighted in point two, the text-to-image task also improves the performance of certain visual tasks.
>
> 4. A key distinction between PixWizard and Emu Edit lies in their open source. Emu Edit doesn’t release any model, and sufficient data curation details are not available in their paper, including how to construct data for different task types, how to identify a unified training paradigm across tasks, data volume and cleaning processes, and model and training specifics. In contrast, PixWizard provides all the details, greatly benefiting both individuals and research institutions in terms of reproducibility and further improvements.

---

> > ### Author Response · Authors · 2024-11-21
> > **Response [2/3] to Official Review by Reviewer kiLh**
> >
> > > **Q2 and W2: What is the extent of generalization that your model actually supports?**
> >
> > Below we provide additional clarification on our model’s generalization:
> >
> > 1. Zero-shot Examples: We provide several zero-shot visual examples in **Appendix. D.6(Fig.15)** of the revised version. These include marking target objects using different symbols other than a box, such as a star, a point, or a circle. Additionally, we can change the color of detection boxes to colors that were not seen during training. Furthermore, we can simultaneously perform multiple tasks, such as deraining and watermark removal, deraining and super-resolution, or inpainting and outpainting, with masks of any shape and mixed black-and-white colors. These examples highlight the generalizability of our approach. However, getting PixWizard to perform entirely new tasks that are completely unrelated to the training data, such as adding an object and then detecting it or contour detection, is challenging. This limitation points to the boundaries of PixWizard's generalization capabilities.
> >
> > 2. Few-shot Tasks: Following Emu Edit's experimental setup, we further validated PixWizard's generalizability from a few-shot learning task. Due to time constraints, we prepared an object contour detection task as an illustration. We constructed 50 training samples for this task (all with red contours) and added them to our training dataset, then fine-tuned PixWizard. As shown in **Appendix. D.6(Fig.15)**, PixWizard quickly acquired this capability. Additionally, we changed the contour color and found that PixWizard could still follow the instructions, implying that the model can effectively generalize to novel tasks.
> >
> > > **W3: Question about ablation study.**
> >
> > We agree that ablation studies are a very important section, and we should better balance different experimental results and visualizations. We have updated and corrected this in the revised paper to help readers gain a clearer understanding of how our new designs operate.
> >
> > > **W4: There is relatively weak motivation for why multi-task models are needed.**
> >
> > We think multi-task models are a direction worth exploring and pursuing deeply. The key reason is that not only can different tasks improve certain tasks' performance, but multi-task models also show great potential in effectively meeting complex demands, especially in industrial scenarios.
> >
> > - Specifically, multi-task models can significantly save memory, storage, and computational resources, simplify the deployment process, and eliminate the need to design complex strategies for switching between different models to accomplish various tasks.
> > - Additionally, multi-task learning better uncovers shared features in data, enhancing model generalization, as demonstrated by various zero-shot generalization performances shown in W2, which are difficult to achieve with single-task models.
> > - More importantly, user interaction with the model becomes more flexible and natural, as illustrated by the interactive methods in Fig.13 of Appendix D.4, which effectively improve results.
> >
> > > **W5: Some components, such as dynamic partitioning and padding schemes, reference recent preprints without citations; if these are important, they should be explained in more detail.**
> >
> > Thank you for pointing this out. We apologize for missing the citation of relevant articles at the first mention of the related concept, and we have updated it in the revised version.
> >
> > While these modules are inherited from the pre-trained model, we agree that providing more details to help readers better understand their mechanisms is necessary. Therefore, we have added relevant details in the revised version in Appendix C.4 to address the concerns of readers and reviewers.

---

> > > ### Author Response · Authors · 2024-11-21
> > > **Response [3/3] to Official Review by Reviewer kiLh**
> > >
> > > > **Q3: What are the details of the MHGS? How is it implemented?**
> > >
> > > In implementation of the Multi-Hot Gumbel-Softmax (MHGS), the specific process is as follows:
> > > - The MHGS function receives input logits, representing the importance of each token. It first adds Gumbel noise through the operation `(logits + GumbelNoise) / temperature`, where the temperature parameter controls the degree of smoothness.
> > > - Next, the noisy logits are normalized using a Softmax function, resulting in a soft output `y_soft`, computed as `y_soft = Softmax(gumbels, dimension=dim)`.
> > > - During training, the model uses `y_soft` to provide smooth gradients for optimization.
> > > - During inference, to produce discrete multi-hot outputs, we applies a `Top-K` operation to select the highest k values, represented by `indices = Top-K(y_soft, k=sample_tokens, dimension=dim)`.
> > > - A hard multi-hot tensor `y_hard` is created using these indices with `y_hard = MultiHotTensor(indices, shape=logits.shape)`.
> > > - The final output ret combines `y_hard` and `y_soft` in the form `ret = y_hard - StopGradient(y_soft) + y_soft`, ensuring discrete values for selected elements while preserving gradient information during backpropagation.
> > >
> > > We have additionally provided pseudocode in Appendix D.7 of the revised version for further explanation.
> > >
> > > > **Q4: Question about multiple resolutions.**
> > >
> > > In Zhuo's paper[1], a separate model needs to be trained for each different scale of resolution. For example, a model based on 512 can only generate images within the range of [512x512, 256x1024, 1024x256...] (with a pixel limit of 512x512 = 262144) but cannot directly generate 1024x1024 images (1024x1024 = 1048576). In contrast, PixWizard supports generating 512x512, 768x768, and 1024x1024 images simultaneously. This is made possible by our grouping strategy for different scale images during training, thereby expanding the range of resolutions and aspect ratios that can be generated.
> > >
> > > [1] Lumina-Next: Making Lumina-T2X Stronger and Faster with Next-DiT
> > >
> > > > **Q5: How to assign manual sampling weights to each dataset**
> > >
> > > Suppose a training dataset contains 100 samples. If the sampling weight is set to 0.5, we randomly select 50 unique samples from this dataset to include in the final training set. Conversely, if the sampling weight is set to 2.0, each sample in this dataset will be duplicated once, resulting in 200 samples in the final training set.
> > > We achieve balanced data distribution by listing the number of samples in each task's training dataset in advance and manually setting the sampling weights for each dataset. This ensures that each task in the final training set has approximately the same amount of data.
> > >
> > > > **Minor Weaknesses Responses.**
> > >
> > > Thank you very much for pointing these out. We have adopted and revised accordingly in the revised paper. Here, we mainly address points 5 and 6.
> > >
> > > For point 5, you did not miss anything. This section was simply intended to highlight one of the advantages of using the flow-matching baseline compared to the previous DDPM-based method.
> > >
> > > For point 6, here is some additional clarification: we tested 100 images with a resolution of 768x768 on an A100 GPU with 60 sampling steps.
> > >
> > > | Method | Infer.(s/img) | Mem.(GB) |
> > > |--|--|--|
> > > | Non-Sparsity| 19.24 | 16.0 |
> > > | Task-aware Dynamic Sampler| 15.68 | 15.5 |
> > >
> > > As shown in the results, fewer tokens provide an improvement in inference and memory efficiency, because we have reduced the computational complexity of cross-attention from quadratic to nearly linear.

---

> > > > ### Comment · Reviewer_kiLh · 2024-11-22
> > > >
> > > > Thank you for your response. I appreciate the detailed and thorough answers, which address my concerns.
> > > >
> > > > Specifically, the list of differences with Emu Edit is fairly convincing, and I suggest that the main differences also be highlighted in the core paper. The finding regarding multi-task training mostly improving related tasks, but possibly diminishing performance on unrelated tasks is interesting and is in my opinion also worth mentioning in the paper.
> > > >
> > > > The novel generalization and few-shot experiments also provide a satisfactory answer to my second primary concern.
> > > >
> > > > Overall, I'm tentatively increasing my score to 6, and will consider raising it further after I read the other reviews/answers and following the reviewer discussion.

---

> > > > > ### Author Response · Authors · 2024-11-23
> > > > > **Thanks for your recognition of our rebuttal!**
> > > > >
> > > > > Dear Reviewer kiLh,
> > > > >
> > > > > Thank you for acknowledging our rebuttal and efforts! We deeply appreciate your insightful comments, which have been invaluable in helping us improve our work.
> > > > >
> > > > > Regards,
> > > > >
> > > > > Authors

---

### Official Review · Reviewer_8hs4 · 2024-10-28

**Soundness:** 3
**Presentation:** 3
**Contribution:** 2
**Rating:** 6
**Confidence:** 4

**Summary:**

The paper proposes a generalist vision model that unifies various image generation and recognition tasks into an image-instruction-to-image problem. To address this problem, they collected a dataset, Omni Pixel-to-Pixel Instruction-tuning Dataset, consisting of 30 million image-instruction-image triplets. Fueled by this dataset, they train a conditional DiT model that takes the conditional image and instruction as the input. The paper shows extensive comparisons against specialized and generalist baselines on various benchmarks.

**Strengths:**

1. Collecting and preprocessing the Omni Pixel-to-Pixel Instruction-tuning Dataset is a huge effort, which is the largest of its kind to my best knowledge and can be a great contribution to the community.

2. The paper is clearly written with rich details.

3. Extensive evaluation is done to demonstrate the performance of the model on various visual tasks, from image editing to image grounding.

4. Ablation studies are performed to verify the proposed methods, helping to understand the significance of each individual components.

**Weaknesses:**

1. While the authors claimed "competitive performance compared to task-specific models and surpass general visual models," this is not directly reflected by the results: there is a clear gap in tasks like depth estimation, semantic segmentation, and image grounding with task-specific models. The general visual model also outperforms some of these tasks. The authors should state the contribution in a more clear way.

2.  The other contribution claimed in the abstract that was not well-demonstrated in the paper is that "the model exhibits promising generalization capabilities with unseen tasks." How much can the model generalize beyond instruction that it does not see during the training? For example, can the model follow the instruction "Mark the specified area with a star in red: {caption}"?

3. The baselines chosen for image generation were not state-of-the-art anymore, e.g., UniControl is not compared, although their dataset (MultiGen-20M) was used.

(Minor) Although some ablation studies are provided to justify the proposed methods, some other choices that seem novel are not well-proved, such as using Gemma-2 instead of T5 as the text encoder and RoPE as the positional embedding.

**Questions:**

1. "To balance tasks, we assign manual sampling weights to each dataset, randomly selecting data when a weight is less than 1.0." - could the author elaborate on how this is done? Like, what does it mean by the weight is less than 1, when the weights are manually assigned?

2. As one of the major contributions of the paper is the public data, will the processed dataset be released in any form?

3. Which layers of the CLIP image features are used as the conditions, is that the 1D feature after pooling or 2D feature before pooling?

**Details Of Ethics Concerns:**

The paper involves a giant dataset, primarily consisting of open-source datasets and self-collected data.

---

> ### Author Response · Authors · 2024-11-21
> **Response [1/1] to Official Review by Reviewer 8hs4**
>
> > **W1: The authors should state the contribution in a clearer way.**
>
> Thank you for pointing this out, and we apologize for inaccurate overclaims. Statistically, among the 12 tasks listed in the experimental table, PixWizard achieved competitive performance with task-specific models in 9 tasks and surpassed general visual models in a majority of tasks. What we intended to convey is that PixWizard demonstrates strong overall multi-task performance. We have updated unclear statements in the revised version.
>
> > **W2: PixWizard's generalization capabilities with unseen tasks.**
>
> We apologize for not providing a detailed explanation in the main paper. Here is some further clarification:
> 1. In Table 4 of the original manuscript, we evaluated tasks that were not included in the training phase, such as under-display camera image restoration and underwater image restoration. PixWizard achieved performance competitive with specialized models, providing initial evidence of its generalizability.
>
> 2. Zero-shot Examples: We provide several zero-shot visual examples in **Appendix. D.6(Fig.15)** of the revised version. These include marking target objects using different symbols other than a box, such as a star, a point, or a circle. Additionally, we can change the color of detection boxes to colors that were not seen during training. Furthcerermore, we can simultaneously perform multiple tasks, such as joint deraining and watermark removal, deraining and super-resolution, or inpainting and outpainting, with masks of any shape and mixed black-and-white colors. These examples highlight the generalizability of our approach. However, getting PixWizard to perform entirely new tasks that are completely unrelated to the training data, such as adding an object and then detecting it or contour detection, is challenging. This limitation points to the boundaries of PixWizard's generalization capabilities.
>
> 3. Few-shot Examples: Following Emu Edit's[1] experimental setup, we further validated PixWizard's generalizability from a few-shot learning perspective. Due to time constraints, we prepared an object contour detection task as an illustration. We constructed 50 training samples for this task (all with red contours), added them to our training dataset, and then fine-tuned PixWizard. As shown in **Appendix. D.6(Fig.15)**, PixWizard quickly acquired this capability. Additionally, we changed the contour color and found that PixWizard could still follow the instructions, implying that the model can effectively generalize to novel tasks.
>
> [1] Emu Edit: Precise Image Editing via Recognition and Generation Tasks
>
> > **W3: The baselines chosen for image generation were not state-of-the-art anymore, e.g., UniControl is not compared.**
>
> Thanks for pointing this out. Here are some additional comparisons:
>
> | |Canny-to-Image |  |  | Depth-to-Image |  |  |
> |-|-:|-:|-:|-:|-:|-:|
> | | F1↑| FID↓| CLIP-S↑| RMSE↓| FID↓| CLIP-S↑|
> |**Uni-ControlNet**|27.32 |17.14 |31.84|40.65 | 20.27 |31.66|
> |**UniControl**| 30.82 |19.94 | 31.97| 39.18 | 18.66 |**32.45**|
> |**PixWizard**| **35.46**| **15.76** | **32.01**| **33.83**| **16.94** |31.84|
>
> > **W4: Some other choices that seem novel are not well-proved, such as using Gemma-2 instead of T5 as the text encoder.**
>
> Since our model was fine-tuned based on the pretrained Lumina-Next-T2I, the original designs, such as Gemma-2 and RoPE, were inherited from the pre-trained model. Modifying these elements would require retraining the pretrained model from scratch, which demands significant resources and time. Therefore, we focused more on exploring designs beyond the pretrained model itself.
>
> > **Q1: Explain the sampling weights to each dataset on training stage.**
>
> Suppose a training dataset contains 100 samples. If the sampling weight is set to 0.5, we randomly select 50 unique samples from this dataset to include in the final training set. Conversely, if the sampling weight is set to 2.0, each sample in this dataset will be duplicated once, resulting in 200 samples in the final training set.
> We achieve balanced data distribution by listing the number of samples in each task's training dataset in advance and manually setting the sampling weights for each dataset. This ensures that each task in the final training set has approximately the same amount of data.
>
> > **Q2: As one of the major contributions of the paper is public data, will the processed dataset be released in any form?**
>
> We promise to make all training and inference codes, model checkpoints, and our processed dataset publicly available. Each data point in the dataset will be a triplet (input image, target image, instruction) to further support the development of the open-source community.
>
> > **Q3: Which layers of the CLIP image features are used as the conditions?**
>
> We use the CLIP image features from the last layer, which is a 2D feature before pooling.
>
> Thank you for your valuable feedback. We sincerely hope our response has addressed your questions.

---

> > ### Comment · Reviewer_8hs4 · 2024-11-24
> >
> > Thank you for your responses and additional experiments. As the additional results support the results, and the authors promise to release the dataset and clarify the contribution in the revised version, I'm happy to increase my rating to 6.

---

> > > ### Author Response · Authors · 2024-11-25
> > > **Thanks for your recognition of our rebuttal!**
> > >
> > > Dear Reviewer 8hs4,
> > >
> > > Thank you for acknowledging our rebuttal and efforts! We deeply appreciate your insightful comments, which have been invaluable in helping us improve our work.
> > >
> > > Regards,
> > >
> > > PixWizard Authors

---

### Official Review · Reviewer_g5qv · 2024-10-29

**Soundness:** 3
**Presentation:** 3
**Contribution:** 2
**Rating:** 6
**Confidence:** 4

**Summary:**

PixWizard presents a unified approach to  conduct various kind of image generation, manipulation, and translation tasks with one DiT model.  The contributions are:

1. The authors constructed an image-instruction-image triplet instruct tuning dataset to train this model.

2. The authors proposed Structural-Aware Guidance to fix the structure of reference image and Semantic-Aware Guidance to learn the instruction-based image to image capability.  Inspired by DynamicVit, the authors designed a task-aware dynamic sampler to squeeze the image tokens for different image to image tasks and utilized Multi-Hot Gumbel-Softmax to extract topK tokens in order to decrease computational cost in DiT.

3. PixWizard inherits the dynamic ability of lumina-next model to handle images of any resolution and aspect ratio. In addition, a two-stage training strategy is designed to improve the performance of the model.

4. Extensive experiments on various dataset and benchmarks concretely verified the generalization of PixWizard

**Strengths:**

1.  The task studied by this paper is the main-stream trend nowadays: unifying multiple image generation tasks in one DiT model.

2.  Solid quantitative experiments demonstrate the generalization capability of the proposed method.

3.  The authors designed a DiT version of DynamicViT. The Multi-Hot Gumbel-Softmax is also interesting. The proposed task-aware dynamic sampler, which sparsifies the image token to reduce computational cost and utilizes one of the two text encoders for task specific embedding, is a brilliant design.

4. Many properties of pixwizard inherit from previous work lumina-next, including the DiT pretrained weigth, dynamic resolutions, which makes pixwizard a meaningful extension of the lumina-next-t2i DiT model in terms of omni image generation.

**Weaknesses:**

1. One of my main concerns is that pixwizard seems to be incapable of conducting non-rigid editing, which is a very common and important, yet challenging topic in image editing, for example, let the standing dog sit, let the polar bear raise its hand, close the open book, etc.  In figure 1, I cannot find  such a common task being presented. I doubt from two perspectives whether pixwizard is capable of conducting such non-rigid editings:

first, the authors designed the structural aware guidance, which concats the vae latent with random noise in  channel dimension. This is identical with instructpix2pix, which claims that "our model is not capable of performing viewpoint changes, can make undesired excessive changes to the image, can sometimes fail to isolate the specified object, and has difficulty reorganizing or swapping objects with each other.".  From my experiments, instructpix2pix completely fails in non-rigid editing. Thus I doubt whether pixwizard inherits such a incapability of instructpix2pix  due to the same concat operation.

second, the datasets utilized for training image editing task are strongly biased and do not contain non-rigid editing data. In section 2, the authors listed the training datasets: UltraEdit (2024), MagicBrush (2024a), GQA-Inpaint (2023), Instruct P2P (2023), SEED-X-Edit (2024), GIER (2020), and HQ-Edit (2024). However, almost none of these datasets contain non-rigid editing data. Instead, these datasets focus on very simple editing tasks, which could simply be solved by inpainting models for adding, deleting and replacing objects or backgrounds,  or controlnet+text to image to change textures and style transfers (of course, pixwizard integrates all these capabilities in one model and demonstrates comparable or better quantitative results than separate models, which is still a contribution. I just want to convey my personal opinion that these tasks are generally considered simple image editing tasks).   These tasks are rather simple in general without changing spatial and structural features. Without training data on non-rigid editing, it is almost impossible for pixwizard to conduct such edits.

2.  my second concern is that although the quantitative experiments are solid, the quantitative results in table 1 show that for many image to image tasks, there is still a significant gap between  pixwizard and task-specific model. In table 2 the image editing results are close, yet the testsets are very biased towards simple editing tasks. In table 3 the results on controlnet and t2i tasks are close though.


3. shown in table 6,  the gain of task aware dynamic sampler and two stage training strategy  is minimal, which weakens the effectiveness of the propose method.

**Questions:**

1. The current state-of-the-art general image editing methods, Imagic, followed by its speedup and less overfitting open-sourced version Forgedit, is capable of conducting general image editing tasks including the non-rigid editing. Is pixwizard capable of conducting non-rigid editing like Imagic and current state-of-the-art  Forgedit? If yes, I would like to see some examples compared with Imagic and  Forgedit with examples from TEdBench in the revised version. If not, I would like to see some discussions on this issue and how to solve it in the revised version.  Imagic and Forgedit require test-time fine-tuning, which costs at least 30 seconds. Is it possible for pixwizard to somehow distill and integrate these methods to tackle the non-rigid editing problem? Considering the time limit of rebuttal period, I won't require the authors to fix this issue in such a short time. Yet  discussions and possible solutions to this problem in the revised paper are compulsory .


2. shown in table 6, the task-aware dynamic sampler do not bring significant performance gain. What are the computational savings then? They are not reported in the paper.

---

> ### Author Response · Authors · 2024-11-21
> **Response [1/1] to Official Review by Reviewer g5qv**
>
> > **Q1 and W1: Could PixWizard perform non-rigid editing? If yes, provide examples and comparisons; if no, provide discussions and possible solutions.**
>
> Thank you for pointing out this critical issue. You are correct; due to the lack of open-source datasets containing non-rigid editing data pairs and the limitations of structural-aware guidance design, the current PixWizard cannot perform non-rigid editing.  We have provided discussions on PixWizard's limitations in the image editing task, along with potential solutions, in our revised paper (Appendix D.5).
>
> > **W2: There is still a significant gap between PixWizard and task-specific model.**
>
> Thank you for pointing this out. It is a meaningful issue to look into. In fact, current generalist vision models cannot comprehensively outperform task-specific models across all sub-tasks, and this is a goal we are striving toward. But we believe that PixWizard offers researchers an important first step: how to unify the multi-task image-to-image paradigm, how to support as many vision tasks as possible, how to standardize the training format across different tasks, and the feasible model structures and training methods. Achieving performance that fully surpasses task-specific models is a long-term goal that we are committed to pursuing persistently.
>
> > **Q2 and W3: The task-aware dynamic sampler does not bring significant performance gain. What are the computational savings?**
>
> The task-aware dynamic sampler was mainly designed to balance the performance of multiple tasks while saving computational costs. We apologize for not providing sufficient explanation in the main paper. Here are some additional experiments: we tested 100 images with a resolution of 768x768 on an A100 GPU with 60 sampling steps:
>
> | Method | Infer.(s/img) | Mem.(GB) |
> |--|--|--|
> | Non-Sparsity| 19.24 | 16.0 |
> | Task-aware Dynamic Sampler| 15.68 | 15.5 |
>
> As shown in the results, although it does not bring a significant performance gain, it offers a notable improvement in inference and memory efficiency.
>
> Thank you for your valuable feedback. We sincerely hope our response has addressed your questions and we would be glad to engage in further discussion if needed.

---

> ### Comment · Reviewer_g5qv · 2024-11-23
> **The authors have stressed my concerns**
>
> The authors have addressed my concerns. I stick to my positive rating for this paper.

---

> > ### Author Response · Authors · 2024-11-24
> > **Thanks for your recognition of our rebuttal!**
> >
> > Dear Reviewer g5qv,
> >
> > Thank you for acknowledging our response and efforts! We greatly appreciate your insightful comments, which have been invaluable in improving our work.
> >
> > Regards,
> >
> > PixWizard Authors

---

### Official Review · Reviewer_Vhoc · 2024-11-04

**Soundness:** 3
**Presentation:** 3
**Contribution:** 2
**Rating:** 6
**Confidence:** 3

**Summary:**

This paper introduces PixWizard, a visual assistant that performs diverse image generation, manipulation, and translation tasks based on free-form language instructions. PixWizard unifies various vision tasks into a single image-text-to-image framework using a newly created Omni Pixel-to-Pixel Instruction-Tuning Dataset. Built on the DiT architecture, PixWizard supports dynamic resolutions, integrates structure- and semantic-aware guidance, and demonstrates competitive performance and generalization across unseen tasks and instructions, making it a powerful interactive tool for visual tasks.

**Strengths:**

- The paper effectively explores the unification of multiple tasks within a single framework.
- The proposed model demonstrates strong performance across a wide range of tasks.
- The method is adaptable to images of varying resolutions.

**Weaknesses:**

The reviewer does not have significant concerns but notes a few minor weaknesses:
- Some implementation details are unclear; For example, in the statement, “In the first stage, we initialize the model by combining the weights of a pretrained text-to-image model with randomly initialized weights for the newly added modules,” it is unclear which model is used for initialization.
- The paper reads more like an engineering report than a research paper with deep insights. However, it would be valuable if the model, code, and/or dataset were made publicly available.
- A concluding analysis on how different tasks influence each other within the unified framework would be beneficial.

**Questions:**

NA

---

> ### Author Response · Authors · 2024-11-21
> **Response [1/1] to Official Review by Reviewer Vhoc**
>
> > **W1: Some implementation details are unclear; For example, it is unclear which model is used for initialization.**
>
> At Line 200 of the original manuscript, we have mentioned that the pretrained model is Lumina-Next-T2I[1].  We have revised the paper to ensure that details are clear.
>
> > **W2: The paper reads more like an engineering report than a research paper with deep insights. However, it would be valuable if the model, code, and/or dataset were made publicly available.**
>
> Thanks for your comments. The proposed framework indeed involves many engineering aspects, especially those related to dataset curation, which are crucial for enhancing reproducibility. We were trying to present details of the framework and make it easy to understand for the general reader.
>
> We promise to make all training and inference code, model checkpoints, and our datasets publicly available to support the further development of the generalist vision models.
>
> > **W3: A concluding analysis on how different tasks influence each other within the unified framework would be beneficial.**
>
> Thanks for the suggestions. Understanding the influences between different tasks is meaningful. Through recent exploration, we conducted a series of experiments. "General" refers to PixWizard trained on multi-task datasets, while "Expert" denotes PixWizard trained solely on single-task datasets. We made the following two findings:
>
> （1）A generalist model does not always outperform the expert model. We found that some tasks, such as image editing and image grounding, show improved performance with multi-task learning, aligning with the experimental results of Emu Edit[2]. However, some vision tasks do not see performance gains from multi-task learning. The example are dense image estimation tasks, as shown in the results below. One possible reason is that their output distributions differ significantly from other tasks, making multi-task learning less beneficial for them.
>
> （2）The text-to-image task can effectively enhance the performance of inpainting, outpainting, and controllable image generation tasks. We believe that this is because these tasks require general image generation capability. Introducing text-to-image strengthens the model's generative ability. However, multi-tasking does not further improve the performance of the text-to-image task.
>
> | |Depth Est. (RMSE↓) |Sem. Seg (mIoU↑) |Derain (SSIM↑) |Grounding (cIoU↑) |Emu Edit (CLIPout↑) |Canny-to-image (FID ↓)|Inpainting (FID ↓)| T2I (HPSv2↑)|
> |--|--|--|--|--|--|--|--|--|
> | **Expert**    | **0.281** | **38.14**| **0.918** | 43.33 | 0.237 | 15.98 |10.32 | 27.47 |
> | **General**   | 0.287 | 36.76 | 0.917 | **46.44**  | **0.248** | **15.66** | **9.27** | 27.47 |
>
> [1] Lumina-Next: Making Lumina-T2X Stronger and Faster with Next-DiT
>
> [2] Emu Edit: Precise Image Editing via Recognition and Generation Tasks
>
> Thank you for your valuable feedback. We sincerely hope our response has addressed your questions and we would be glad to engage in further discussion if needed.

---

### Meta-Review · Area_Chair_epTn · 2024-12-16

**Metareview:**

All the 4 reviewers provide positive ratings after rebuttal, with 2 upgraded score. Initially, the reviewers had concerns about technical contributions, performance gains compared to task-specific models and prior work like Emu Edit, and generalization ability to various instructions. In the post-rebuttal discussion period, all the reviewers are satisfactory with the authors' comments and revised paper. After taking a close look at the paper, rebuttal, and discussions, the AC agrees with reviewers' feedback of the proposed method being effective to perform multiple tasks within a single framework. Therefore, the AC recommends the acceptance rating.

Moreover, although the paper may look like an engineering framework with massive data collection (as reviewer Vhoc pointed out), the AC recognizes the contribution of having a single model for multiple image generation/editing tasks with comprehensive evaluation results. The AC would strongly recommend authors to have a proper release of dataset, model, and training code to benefit the wider community.

**Additional Comments On Reviewer Discussion:**

Initially, the reviewer g5qv, 8hs4, and kiLh had major concerns about the performance comparisons with task-specific models and prior work (e.g., Emu Edit and UniControl). During the post-rebuttal discussion period, the authors have provided more explanations and results to validate that the proposed method is able to achieve competitive results. The AC considers all the feedback and agrees with reviewers' assessment.

---

### Decision · Program_Chairs · 2025-01-22

Accept (Poster)